



# Neodymium budget in the Mediterranean Sea: Evaluating the role of atmospheric dusts using a high-resolution dynamical-biogeochemical model

Mohamed Ayache[1], Jean-Claude Dutay[1], Kazuyo Tachikawa[2], Thomas Arsouze[3], and Catherine Jeandel[4]

[1]Laboratoire des Sciences du Climat et de l'Environnement, CEA-CNRS- Université Paris Saclay, Gif-sur-Yvette, France
[2]Aix Marseille Univ, CNRS, IRD, INRAE, Coll France, CEREGE, Aix-en-Provence, 13545, Aix-en-Provence, France
[3]Barcelona Supercomputing Center, Barcelona, 08034, Spain
[4]LEGOS, University of Toulouse, CNRS, CNES, IRD, UPS, Toulouse, 31400, France

**Correspondence:** Mohamed Ayache (mohamed.ayache@lsce.ipsl.fr)

**Abstract.** The relative importance of river solid discharge, deposited sediment remobilisation and atmospheric dust as sources of neodymium (Nd) to the ocean is the subject of ongoing debate, the magnitudes of these fluxes being associated with a significant uncertainty. The Mediterranean basin is a specific basin; it receives a vast amount of emissions from different sources and is surrounded by continental margins, with a significant input of dust as compared to the global ocean. Furthermore, it is

largely impacted by the Atlantic water inflow via the Strait of Gibraltar. Here, we present the first simulation of dissolved Nd concentration ([Nd]) and Nd isotopic composition ($\varepsilon$Nd) in the Mediterranean Sea using a high-resolution regional model (NEMO/MED12/PISCES) with an explicit representation of all Nd inputs, and the internal cycle, *i.e.* the interactions between the particulate and dissolved phases. The high resolution of the oceanic model (at 1/12°), essential to the simulation of a realistic Mediterranean circulation in present-day conditions, gives a unique opportunity to better apprehend the processes governing

the Nd distribution in the marine environment. The model succeeds in simulating the main features of $\varepsilon$Nd and produces a realistic distribution of [Nd] in the Mediterranean Sea. We estimated the boundary exchange (BE, which represents the transfer of elements from the margin to the sea and their removal by scavenging) flux at $89.43 \times 10^6$ g(Nd)/yr, representing $\sim$90 % of the total external Nd source to the Mediterranean basin. The river discharge provided $3.66 \times 10^6$ g(Nd)/yr, or 3.7 % of the total Nd flow into the Mediterranean. The flux of Nd from partially dissolved atmospheric dusts was estimated at $5.2 \times 10^6$

g(Nd)/yr, representing 5.3 % of the total Nd input. This work highlights that the impact of river discharge on [Nd] is localized near the catchments of the main rivers. In contrast, the atmospheric dust input has a basin-wide influence, correcting for a too-radiogenic $\varepsilon$Nd when only the BE input is considered, and improving the agreement of simulated dissolved Nd concentration with field data. This work also suggests that $\varepsilon$Nd is sensitive to the spatial distribution of Nd in the atmospheric dust, and that the parametrisation of the vertical cycling (scavenging/remineralisation) considerably constrains the ability of the model to

simulate the vertical profile of $\varepsilon$Nd.



# 1 Introduction

The Nd isotopic composition ($\varepsilon$Nd) is one of the most useful tracers to fingerprint water mass provenance (see Tachikawa et al. 2017, for a review). Substantial progress has been made during the last few decades in our knowledge of processes/mechanisms

controlling the Nd oceanic cycle, through coordinated high-quality sampling and measurements (*e.g.* GEOTRACES program) and modeling efforts (*e.g.* (Tachikawa et al., 2003; Arsouze et al., 2007; Siddall et al., 2008; Arsouze et al., 2009; Jones et al., 2008; Rempfer et al., 2011). However, the use of $\varepsilon$Nd as a water mass tracer is hampered by the lack of adequate quantification of the external sources, including inputs from river discharge, atmospheric dusts, benthic fluxes, submarine ground water discharge, hydro-thermal sources, and exchange with the sediments at the continental margins (Fig. 1) (*e.g.,*

Goldstein and O'nions, 1981; Piepgras and Wasserburg, 1987; Frank, 2002; Goldstein and Hemming, 2003; Lacan and Jeandel, 2005; Johannesson and Burdige, 2007; Abbott et al., 2015; Morrison et al., 2019; Pöppelmeier et al., 2019).

The Mediterranean basin provides an excellent opportunity to improve our understating of the Nd oceanic cycle and further develop the existing modeling approach. The Mediterranean Sea is a semi-enclosed basin with a relatively short water residence time ($\sim$ 100 years; Millot and Taupier-Letage, 2005), receiving vast amounts of inputs from various sources. It is strongly

connected to continental margins, with a coastline of more than 45 000 km and significant freshwater inputs compared with the open ocean (Ludwig et al., 2009; Ayache et al., 2020). Many studies have shown that dust deposition from the Sahara and Middle East is a significant source of dissolved trace elements to the upper layers of the Mediterranean Sea (*e.g.,* Dulac et al., 1989; Guieu et al., 2002; Richon et al., 2018). The impacts of dust deposition on the Nd distribution are not fully understood and may change in the future as a result of the effects of climate change on land and sea (*e.g.,* Peñuelas et al., 2013). The vertical

profile of dissolved Nd in the Mediterranean Sea is atypical, with a high concentration in the surface water that suggests a significant impact of external sources. Thus, the Mediterranean Sea is ideal to examine the influence of external sources of Nd versus that of the internal cycle (*i.e.* scavenging/remineralisation). Recently, the Meteor and MedBlack/GEOTRACES projects have led to a large increase in the number of observations of Nd in the Mediterranean basin (Tachikawa et al., 2004; Garcia-Solsona and Jeandel, 2020; Montagna et al., 2022). These authors have shown that seawater $\varepsilon$Nd values behave overall

conservatively in the open Mediterranean Sea and confirmed that water masses are distinguishable by their Nd isotope signature (Tachikawa et al., 2004; Montagna et al., 2022). This data provides a unique opportunity to test models describing the cycling of Nd in the Mediterranean Sea. Modeling represents an interesting approach to investigate the impact of external inputs on the oceanic Nd cycle, and we dispose of a high spatial resolution regional model (NEMO-MED12), essential to the simulation of a realistic Mediterranean Sea circulation.

Many modeling studies contributed to improve our understanding of the Nd oceanic cycle. Arsouze et al. (2007) highlighted the importance of boundary exchange (BE) as a source/sink of Nd; however, in their simplified preliminary study, they neglected the Nd inputs from river and atmospheric dust. Jones et al. (2008) used in-situ observations to prescribe surface $\varepsilon$Nd (i.e., they considered no external inputs) and $\varepsilon$Nd as a quasi-conservative of mixing at global scale. Siddall et al. (2008) explicitly simulated the [Nd] and $\varepsilon$Nd using a reversible-scavenging model and fixed surface boundary conditions. They concluded

that reversible scavenging is an active and important component in the cycling of Nd and should be considered a necessary



component in explaining the Nd paradox[1]. Arsouze et al. (2009) simulated [Nd] and $\varepsilon$Nd simultaneously using a fully coupled dynamical/biogeochemical model and a reversible scavenging model. They also explicitly represented the BE, river, and dust deposition Nd sources. Their study confirmed that sediment dissolution is the main Nd source to the oceanic reservoir, representing 95% of the total Nd source, with the associated boundary scavenging process representing up to 64% of the total Nd

sink. In this global study, river discharge ($2.6 \times 10^8$ g(Nd)/yr) and dust atmospheric inputs ($1.0 \times 10^8$ g(Nd)/yr) are significantly lower than the Nd BE inputs. Using a similar approach, Gu et al. (2019) assessed the response of the Nd cycle to freshwater forcing. Ayache et al. (2021) used the simplified version of the $\varepsilon$Nd simulation proposed by Arsouze et al. (2007) to investigate with idealized hosing experiments in the IPSL-CM5 model the link between the intensification of the upper AMOC (Atlantic meridional overturning circulation) and the Mediterranean overflow. Pöppelmeier et al. (2019) used the Nd-enabled Bern3D

model, and included a parametrization of the benthic Nd source extended over all water depths. This study suggested that the contributions of the Nd sources are ~60% boundary/benthic source, ~32% riverine source, and ~9% dust; however, the coarse resolution of this model limits its ability to sufficiently resolve the processes affecting the Nd oceanic cycle.

A large proportion of BE is thought to occur predominantly within estuarine sediments and on continental margins (Rousseau et al., 2015). Thus, the dissolution of only a small proportion (1–3 %) of particulate material deposited within estuarine

sediments and on continental margins can have a large impact on marine Nd budgets and cycling (Jeandel and Oelkers, 2015). Arsouze et al. (2007) suggested that the BE rate is poorly sensitive to the lithology of the margin sediments (*e.g.* granitic vs. basaltic). Nevertheless, it is important to mention that the magnitudes and variations of Nd fluxes related to the partial dissolution of river particles and atmospheric dust bear a significant uncertainty because the estimated dissolution rates of Nd from dust vary from 2 to 50 % (Tachikawa et al., 1999; Greaves et al., 1994; van de Flierdt et al., 2004). Nd concentration in

the river discharge is generally prescribed in modelling experiments using a subtraction percentage of solid material, which varies from 30 % (Rempfer et al., 2011; Gu et al., 2019) to 70 % (Arsouze et al., 2009; Nozaki and Zhang, 1995; Sholkovitz et al., 1994; Elderfield et al., 1990).

The Nd influx brought by the Atlantic inflow through the Strait of Gibraltar is smaller than the Nd outflux exiting with the Mediterranean outflow (Tachikawa et al., 2004; Henry et al., 1994; Greaves et al., 1991), and the $\varepsilon$Nd value of the Mediter-

ranean outflow ($\varepsilon$Nd = -9.4, Tachikawa et al.,; 2004) is higher than that of Atlantic inflow water ($\varepsilon$Nd = -11.8; Spivack and Wasserburg, 1988). Thus, a source of radiogenic Nd in the Mediterranean Sea is required to balance these fluxes. The first Nd budget for the Mediterranean Sea was proposed by Frost et al. (1986) and Spivack and Wasserburg (1988). These authors suggested that the additional Nd source could be river particles and/or dust particles. Greaves et al. (1991), using the Rare Earth Elements (REE) patterns of seawater, argued that the missing source might rather be of marine origin. Schijf et al. (1991) pro-

posed that the Black Sea was a net source to the Mediterranean Sea. Based on a two-box model, Henry et al. (1994) highlighted that the $\varepsilon$Nd in the North West deep waters required an exchange involving $30 \pm 20$ % of the sinking particles of atmospheric origin. More recently, Tachikawa et al. (2004) proposed that the missing term could be partially dissolved Nile river particles. Montagna et al. (2022) suggest that the relative importance of dust in modifying the $\varepsilon$Nd signature of surface waters in the

---

[1]Nd paradox: define decoupling of $\varepsilon$Nd [Nd] in the water column, *i.e.* $\varepsilon$Nd behave quasi-conservatively, while [Nd] in the water column generally increase with depth, showing a broadly nutrient-like behavior (Tachikawa et al., 2003; Goldstein and Hemming, 2003; Lacan and Jeandel, 2005)





Mediterranean Sea is minor, and they associate the very radiogenic $\varepsilon$Nd signature in the Leventine sub-basin to the dispersion

of Nile river particles in the surface layer. However, the Nile water discharge was drastically reduced after the construction of the Aswan High Dam in 1964. Furthermore, the few existing estimations of Nd in atmospheric dusts are based on local observations that are not necessarily representative of the whole basin. River inputs and water exchange with the Black sea (via the Dardanelles strait) are still not fully constrained.

Ayache et al. (2016) proposed the first simulation of $\varepsilon$Nd using a regional high-resolution dynamical model (at 1/12° of

horizontal resolution) of the Mediterranean including only BE, and using a relaxing term applied to the first 540 m of the continental margin. Their work confirms previous findings that boundary exchange is a major process in the Nd oceanic cycle, even at the regional scale and in a semi-enclosed basin such as the Mediterranean basin. Nevertheless, this simplified approach simulated too-radiogenic $\varepsilon$Nd values in the Mediterranean Sea, and did not represent the Nd inputs from river and atmospheric dust. In the present study, we extend this Nd cycle modeling effort in the Mediterranean Sea by simulating both

$\varepsilon$Nd and the Nd concentration following the protocol proposed by Arsouze et al. (2009) for the global ocean. We use a high-resolution regional model with an explicit representation of all Nd sources (*i.e.* margin sediment re-dissolution, dissolved river fluxes, and atmospheric dusts) and sinks (*i.e.* scavenging). Vertical cycling is simulated using a reversible scavenging model developed for the simulation of trace elements using the biogeochemical circulation model NEMO–PISCES (Dutay et al., 2009; Arsouze et al., 2009; Hulten et al., 2018). We performed several sensitivity tests to better understand how the internal cycle

(scavenging/remineralisation) and the various external sources affect the Nd cycle in the Mediterranean Sea, and particularly assess how it is impacted by atmospheric inputs in this region, where desert dust deposition events are more frequent affecting a large spatial domain compared to the global ocean.

## 2 Methods

### 2.1 Circulation via NEMO-MED12 model

The dynamical model used in this work is the NEMO (Nucleus for European Modelling of the Ocean) free surface ocean general circulation model (Madec and NEMO-Team., 2008) in a regional high-resolution configuration (at 1/12° = ∼ 7 km) called NEMO-MED12 (Beuvier et al., 2012a). The NEMO-MED12 domain covers the whole Mediterranean Sea and includes part of the Atlantic Ocean west of Gibraltar (buffer zone) from 30-47°N in latitude and from 11°W-36°E in longitude, where temperature and salinity (3-D fields) are relaxed to the observed climatology (Beuvier et al., 2012b). Water exchange with the

Black Sea is represented as a two-layer flow with net budget estimates from Stanev and Peneva (2002).

The NEMO-MED12 model is forced at the surface by the momentum, evaporation and heat fluxes over the period 1958–2013 from the ARPERA model (Herrmann and Somot, 2008; Herrmann et al., 2010). The sea-surface temperature (SST) and water-flux correction term are applied using ERA-40 (Beuvier et al., 2012b). River and runoff discharge are derived from the model of Ludwig et al. (2009) and the inter-annual data set of Vörösmarty et al. (1996). The initial conditions (salinity and temperature)

are provided by the MedAtlas-II (MEDAR-MedAtlas-group, 2002; Rixen et al., 2005). The initial state in the buffer zone is prescribed from the World Ocean Atlas 2005 (Locarnini et al., 2006; Antonov et al., 2006). The sea-surface height (SSH) is





restored in the buffer zone toward the GLORYS1 reanalysis (Ferry et al., 2010) in order to conserve the total volume of water in the Mediterranean Sea.

The NEMO-MED12 model has been used previously for many oceanic investigations in the Mediterranean Sea (*e.g.,* Brossier et al., 2011; Beuvier et al., 2012b; Soto-Navarro et al., 2014; Ayache et al., 2015a, b, 2016, 2017; Palmiéri et al., 2015; Guyennon et al., 2015; Richon et al., 2018). The NEMO-MED12 model simulates the main structures of the thermohaline circulation of the Mediterranean Sea, with mechanisms having a realistic timescale compared to observations (Ayache et al., 2015a). However, some aspects of the model still need to be improved: for instance, the too weak formation of Adriatic Deep Water (AdDW) as shown by Ayache et al. (2015a) using anthropogenic tritium simulations. In the western basin, the production of WMDW is well reproduced, but the spreading of the recently ventilated deep water to the south of the basin is too weak (Ayache et al., 2015a). Full details of the model and its parametrizations are described by Beuvier et al. (2012a, b); Palmiéri et al. (2015); Ayache et al. (2015a).

## 2.2 Particle dynamics via PISCES model

The biogeochemical model PISCES (Aumont and Bopp, 2006; Aumont et al., 2015) is coupled to the regional physical model NEMO-MED12 (Palmiéri, 2014; Richon et al., 2018). PISCES simulates the biogeochemical cycles of carbon, oxygen and five nutrients (nitrates, phosphates, ammonium, silicates and iron) that can limit phytoplankton growth. It explicitly simulates two trophic levels: phytoplankton groups (nanophytoplankton and diatoms) and zooplankton groups (microzooplankton and mesozooplankton). PISCES is a Redfieldian model where the C/N/P ratio used for plankton growth is fixed to 122/16/1.

There are three non-living compartments simulated by PISCES: dissolved organic carbon (DOC), large particles and small particles, the latter two differing by their sinking velocities. The large particle pool includes: particulate organic carbon with a diameter larger than 100 $\mu$m ($POC_b$), biogenic silica (BSi), carbonate ($CaCO_3$) and lithogenic particles (atmospheric dust), sinking with a velocity of 50 m/day. Small particles consist of particulate organic carbon between 2 and 100 $\mu$m in size ($POC_s$) and a sinking velocity of 3 m/day. The small particle pool represents the principal stock of particles at the surface (Dutay et al., 2009). The content of the particulate pools is controlled by mineralization, mortality, grazing, and the two POC classes interact via the processes of disaggregation and aggregation (see Aumont and Bopp, 2006 and Dutay et al., 2009). We use PISCES in its offline mode, where biogeochemical tracers are transported using an advection–diffusion scheme driven by dynamical variables (velocities, pressure, mixing coefficients) previously calculated by the oceanic model NEMO-MED12 (Palmiéri et al., 2015).

## 2.3 The reversible scavenging model

The observation indicate that Nd concentrations generally increase in the ocean with depth (Baar et al., 1985) as a consequence of a continuous and reversible exchange between the particulate and dissolved phases (Nozaki and Alibo, 2003). This process is called the reversible scavenging, *i.e.* isotope adsorption into sinking particles in the surface and is redissolved at depth. The equilibrium scavenging approach is commonly used in Nd and Pa/Th modelling (Siddall et al., 2005; Dutay et al., 2009; Arsouze et al., 2009; Gu and Liu, 2017; Hulten et al., 2018). It allows the model to use partition coefficients that can be directly





constrained by observations (although consensus values for these coefficients are still not available). This approach considers that the partition between dissolved and particulate phase is in equilibrium, as suggested by observations (*e.g.,* Roy-Barman et al., 1996), and their relative contribution is set using an equilibrium partition coefficient K, defined as:.

$$K = \frac{Nd_p}{Nd_p C_p} \tag{1}$$

where $C_p$ is the mass of particles per mass of water. This coefficient K is defined for each type of particles represented in

the model: big ($POC_b$) and small ($POC_s$) Particulate Organic Carbon, calcite ($CaCO_3$), Biogenic Silica (BSi), and lithogenic atmospheric dust (litho). Flowing Arsouze et al. (2009) we simulate the two $^{144}$Nd and $^{143}$Nd isotopes independently (simulates as two tracers) then we calculate total Nd concentration and $\varepsilon_{Nd}$. In-situ observation do not suggest any fractionation between this two isotopes of Nd (*i.e.* $^{144}$Nd and $^{143}$Nd), and their masses are quite similar (Dahlqvist et al., 2005). Hence, partition coefficients (K) are thus assumed as being identical for the two isotopes for each particle type (Arsouze et al., 2009).

The total concentration ($Nd_T$), defined as the sum of big ($Nd_{pg}$: $POC_b$, $CaCO_3$, BSi, litho), small ($Nd_{ps}$: $POC_s$) particulate concentration, and dissolved concentration ($Nd_d$).

$$Nd_T = Nd_{ps} + Nd_{pb} + Nd_d \tag{2}$$

Applying Eq. (1) to the particulate pools in Eq. (2) to express total concentration as a function of dissolved Nd concentration, we obtain:

$$Nd_T = (K_{POCs} * C_{POCs} + K_{POCb} * C_{POCb} + K_{BSi} * C_{BSi} + K_{CaCO3} * C_{CaCO3} + K_{litho} * C_{litho} + 1) * Nd_d \tag{3}$$

From that we can calculate the Nd in small particulate concentration by (equation 4):

$$Nd_{ps} = \frac{K_{POCs} * C_{POCs}}{K_{POCs} * C_{POCs} + K_{POCb} * C_{POCb} + K_{BSi} * C_{BSi} + K_{CaCO3} * C_{CaCO3} + K_{litho} * C_{litho} + 1} * Nd_T \tag{4}$$

the Nd in big particulate concentration by (equation 5):

$$Nd_{pb} = \frac{K_{POCb} * C_{POCb} + K_{BSi} * C_{BSi} + K_{CaCO3} * C_{CaCO3} + K_{litho} * C_{litho}}{K_{POCs} * C_{POCs} + K_{POCb} * C_{POCb} + K_{BSi} * C_{BSi} + K_{CaCO3} * C_{CaCO3} + K_{litho} * C_{litho} + 1} * Nd_T \tag{5}$$


This approach allows to define the [Nd] in big and small particles as a function of the total Nd concentration ($Nd_T$) and partition coefficients (K). This method confers a great advantage in that only the two isotope of Nd ($^{144}$Nd and $^{143}$Nd) are



transported by the model, rather than concentration in every phase (all big particles, small particles and dissolved phase, *i.e.* 12

tracers), which implies a substantial gain of computational cost.

The evolution of the simulated total Nd concentration ($Nd_T$) is equal to the sum of all sources of Nd, impact of vertical cycling (equation 6) and the three-dimensional advection and diffusion (*i.e.* physical transport).

$$\frac{\delta Nd_T}{\delta t} = \overbrace{S(Nd_T)}^{\text{(Source of Nd)}} - \overbrace{\frac{\delta(\omega_s Nd_{ps})}{\delta z} + \frac{\delta(\omega_b Nd_{pb})}{\delta z}}^{\text{(Vertical cycling)}} \overbrace{-U \cdot \nabla Nd_T + \nabla \cdot (K \nabla Nd_T)}^{\text{(3-D advection and diffusion)}} \tag{6}$$

where S(NdT) represents the Source term of the Nd in the model (*cf.* Sect. 2.4).

The vertical cycling represents the scavenging of Nd by the the big and small particles ($\omega_s$ and $\omega_b$ are the sinking velocities of small and big particles, respectively, *cf.* Table 1). Moreover the simulations are performed in off-line mode using the pre-computed transport fields and particles fields ($POC_s$, $POC_b$, $CaCO_3$ and $BSi$) at the monthly time scale. This method requires considerably lower computational cost which allowed to run a relatively long simulation with a high resolution regional model, and to perform some sensitivity tests on Nd values in atmospheric dusts and the values of the partition coefficients.

## 2.4   External inputs and boundary conditions of Nd

There is an ongoing debate on the Nd inputs to the ocean (see Sec. 1). Ayache et al. (2016) established a new map of published [Nd] and $\varepsilon$Nd for the whole Mediterranean basin, based on various types of samples: river discharge, sedimentary material, and/or geological material outcropping above or close to a margin. We therefore use this database to explicitly represent the various sources of Nd in the Mediterranean Sea.

The BE source is implemented in the model as the Nd input from sediment remobilization following the parametrization proposed by Arsouze et al. (2009). This source is imposed in the model as an input flux (S(NdT)sed, *cf.* equation 7) for each grid point of the continental shelf:

$$S(Nd_T)sed = \int_S F_{sed} * mask_{margin} \tag{7}$$

where $F_{sed}$ is the source flux of sedimentary Nd to the ocean and $mask_{margin}$ is the percentage of continental margin in

the grid box and represents the proportion of the surface in the grid where the BE process occurs. We computed $F_{sed}$ for both $^{144}$Nd and $^{143}$Nd by using the Nd concentration and the isotopic composition along the margin presented in figure 2 (Fig. 2a and 2b, see Sect. 1). The oceanic margin extension of the Mediterranean Sea has been chosen to be between 0 and ∼540 m following the margin definition used to model the biogeochemical cycles in the Mediterranean Sea by Palmiéri (2014). To date, there is no estimation of the Nd flux from the sediment (*i.e.* the Boundary Source) in the Mediterranean Sea. Based on our

modeling approach, we estimate the BE flux at 89.43 ×10$^6$ g(Nd)/yr for the whole Mediterranean basin (as presented above, see Table. 2). We compared the compilation of [Nd] (Fig. 2a) and $\varepsilon$Nd (Fig. 2b) along the Mediterranean continental margin proposed by Ayache et al. (2016) with the new global database of Nd provided by Blanchet (2019) and the recent update of





global continental and marine Nd by Robinson et al. (2021). Margin Nd isotopic signatures vary from radiogenic values (up to +6) in the Aegean and Levantine sub-basins to non-radiogenic values in the Gulf of Lion, (∼-11), and [Nd] globally varies

from low [Nd] in the western basin (∼25 $\mu$g/g) to a higher [Nd] in the southeastern basin (∼40 $\mu$g/g). The two maps of $\varepsilon$Nd and [Nd] provided by Ayache et al. (2016) are in good agreement with the new database (Robinson et al., 2021), except on the Libyan coast where the new update suggests a less radiogenic $\varepsilon$Nd (∼-13) and a relatively lower [Nd] of ∼35 $\mu$g/g (Robinson et al., 2021).

In addition to the sediment remobilization source, we implemented the Nd inputs from river/runoff discharge and atmo-

spheric dusts deposition in surface waters as follows:

$$S(Nd_T)surf = \int\limits_S F_{surf} \tag{8}$$

where $F_{surf}$ is the Nd flux from river discharge and atmospheric dusts (in g(Nd)/m2/yr) as presented in Fig. 2.

River discharge is derived from the inter-annual data sets of Ludwig et al. (2009) and Vörösmarty et al. (1996), and we used the runoff estimation provided by the NEMO-MED12 model in Beuvier et al. (2010, 2012b) and Palmiéri et al. (2015). [Nd]

(Fig. 2c) and $\varepsilon$Nd (Fig. 2d) in river inputs are from Ayache et al. (2016). As for margin data, we compared the [Nd] and $\varepsilon$Nd from river discharge/runoff (Ayache et al., 2016) against the new database of Blanchet (2019). The main river systems of the Mediterranean basin are the Nile, Po and Rhone. The Nile river is the largest source of radiogenic Nd to the eastern basin as suggested by Tachikawa et al. (2004). The Rhone river accounts for most of the riverine discharge in the northwestern basin. Based on the runoff estimation of the NEMO-MED12 model, we obtain a dissolved Nd flux from river waters of $3.66 \times 10^6$

g(Nd)/yr (see Table 3).

Atmospheric deposition forcing of dust is provided by the monthly maps from the ALADIN-Climate model (Nabat et al., 2015) used by (Richon et al., 2018) to simulate the impacts of atmospheric deposition of nitrogen and desert dust-derived phosphorus on the biological budgets of the Mediterranean Sea (Fig. 2g). $\varepsilon$Nd values were extracted from Sheuvens et Blanchet (Fig. 2e and 2f). In the areas where no data were available, $\varepsilon$Nd and [Nd] of the atmospheric dust were determined based on the

average values estimated by Tachikawa et al. (2004) for African dust and the value for the region of origin of the dusts provided by Scheuvens et al. (2013). The regional distribution of the Nd values shows that these values are relatively high (∼-9.2) in the eastern part of northern Africa (*e.g.*, Egypt), compared with the central and western parts of northern Africa, where Nd ranges from -17.9 to -11.8 (Scheuvens et al., 2013). Atmospheric dust deposits are taken into account as Nd inputs in surface waters (first vertical level). As the Nd solubility is uncertain, we performed many sensitivity test simulations on the dissolution

rates of particulate Nd from atmospheric dusts, and on the spatial distribution of Nd concentration and isotopic composition in atmospheric dust (*cf.* section 2.5).

## 2.5 Simulations and sensitivity tests

The main objective of this study was to identify and quantify the various sources involved in the Nd cycle in the Mediterranean Sea. With this aim, five distinct simulations were performed (SedOnly, SedRiv, SedRivDust, Dust-CST and Dust-EWbasin,





Table 2). The SedOnly experience considered sediment remobilization as the unique source of Nd. The SedRiv simulation
considered dissolved river discharge in addition to sediment remobilization. In the SedRivDust simulation, we explicitly represented the three main inputs of Nd (*i.e.* sediment remobilization, river discharge, and atmospheric dust). In order to explore
the sensitivity of simulated Mediterranean water Nd concentration and isotopic composition to the spatial distribution of the
atmospheric Nd flux, we performed two more simulations under different dust supplies. In the Dust-CST simulation, the con-

ditions were the same as in SedRivDust except that $\varepsilon$Nd and [Nd] in atmospheric dusts were set constant at -12 and 30 $\mu$g/g,
respectively (estimated as the average values of previously published data over the whole Mediterranean basin). In the Dust-
EWbasin simulation, we applied constant values of $\varepsilon$Nd and [Nd] in each basin (*i.e.* average value for each basin): Nd= -11
and [Nd]= 31 $\mu$g/g in the eastern basin, and Nd= -12.5 and [Nd] =27.5 $\mu$g/g in the western basin.

    Yet, significant uncertainty remains about the dissolution rates of particulate Nd from atmospheric dusts, which are suggested

to vary from 2 to 50 % (*e.g.,* Greaves et al., 1994; Tachikawa et al., 1999). More recently, Zhang (2008) estimated that this
percentage does not exceed 10%. Arsouze et al. (2009) and Gu et al. (2019) used a ratio of 2 % for the global Nd budget, *i.e.*
only 2 % of Nd brought by aeolian dusts are dissolved by contact with seawater and 98 % sinks directly with the particles to
the seafloor. Arsouze et al. (2009) performed sensitivity tests on the dissolution rate of Nd in atmospheric dusts, which did not
significantly change the results of the simulation at the global scale. In order to examine the impact of greater dust dissolution

on the Nd distribution at the regional scale, we performed an additional simulation in which we increased the Nd dissolution
rate in the atmospheric dusts from 2 to 10 % (*i.e.* the maximum value as suggested by Zhang, 2008).

    In the present study, we use equilibrium partition coefficients, "K," from previous modeling studies (Arsouze et al., 2009;
Rempfer et al., 2011; Gu et al., 2019). However, the K values of the partition coefficients are still difficult to constrain because
still of the very limited data are available and because all the modeling studies were made at the global scale. We first used the

partition coefficient previously considered in previous Nd modelling studies in the global ocean with the PISCES model, and
we performed some sensitivity simulations on the K values (see section 4.3 and Fig. A1 and Fig. A2 in appendix).

## 3   Results

### 3.1   Nd concentration

    In order to explore the impact of each source of Nd to the Mediterranean Sea, we performed a series of simulations sequentially

integrating the various external sources: SedOnly, SedRiv, and SedRivDust. The resulting horizontal distributions of [Nd] in
the surface (0–200 m), intermediate (200–600 m), and deep waters (600–3500 m) are represented in Fig. 3, together with a
compilation of in situ observations from Spivack and Wasserburg (1988), Greaves et al. (1991), Tachikawa et al. (2004), Vance
et al. (2004), Henry et al. (1994), Dubois-Dauphin et al. (2017), Gacic et al. (2010), and Montagna et al. (2022).

    Figure 4 shows the Nd concentrations along a longitudinal transect in both the eastern (EMed) and western (WMed) basins

for the three experiences. Without atmospheric dust (SedOnly and SedRiv), the simulated Nd concentrations are globally
similar, homogeneous and very low compared to observations in the whole water column (Figs. 3, 4). More particularly, in
surface waters, the simulated Nd concentrations are lower than 4 pmol/kg while observations indicate values of ∼30 pmol/kg



(Tachikawa et al., 2004). Nd concentration is increasing with depth in these two experiments, however, simulated concentrations only amount to roughly half the observed concentrations in intermediate and deep waters. Adding atmospheric deposition
in the SedRivDust experience considerably enhances Nd concentrations and improves the modeling results. Simulated [Nd] are increasing in the whole water column, towards levels similar to the observations (Fig. 3g, h, and i). However, the surface layer Nd concentration increase leads to values up to 10 pmol/kg in the western basin and of the order of 25 pmol/kg on average in the eastern basin (Fig. 4e); these values are more comparable to but still lower than the observations, of 30 pmol/kg on average in the whole basin. Including the atmospheric dust inputs in the SedRivDust experience also changes drastically the
vertical distribution of the tracer. It is particularly well illustrated by the averaged vertical profiles against in-situ observations constructed in the eastern and western basin and the whole Mediterranean Sea (Fig. 5). The consideration of atmospheric dust inputs generates a more realistic vertical profile and produces a Nd concentration maximum in the subsurface layer (200–800 m depth) detected in the observations that was not simulated in the first two experiments. The two experiments, with a constant [Nd] value for the whole basin (Dust-CST) and with constant [Nd] values for each basin (Dust-EWbasin), lead to relatively
similar results for [Nd] in the surface water and average [Nd] vertical profiles to those of the SedRivDust experience, as shown in Fig. 5.

## 3.2   Isotopic composition

In surface waters, the three experiments generate an E-W gradient in $\varepsilon$Nd, with more radiogenic values in the eastern basin than in the western basin. This is consistent with the observations (Fig. 6). However, the first two simulations, SedOnly and
SedRiv, globally overestimate the surface isotopic signatures with unrealistic radiogenic values in the Aegean Sea and around the Egyptian coast. In intermediate and deep waters, modeled Nd isotopic composition values are globally in agreement with the observations in the eastern basin (Figs. 5, 6, and 7) but largely too radiogenic in the western basin. Considering atmospheric deposition (SedRivDust) again significantly improves the results, producing more realistic Nd isotopic signatures in the surface water of the western basin (Fig. 6g).
The SedRivDust model simulates the observed $\varepsilon$Nd east–west gradient characterizing the surface waters (Fig. 6a), with unradiogenic waters from the Atlantic progressively shifting toward more radiogenic values in the Levantine basin (Tachikawa et al., 2004). The extrema in the Aegean sub-basin and along the Egyptian coast that are simulated in SedOnly and SedRiv are reduced toward more realistic values. The modelled isotopic composition now reproduces more correctly the observed E–W gradients in the intermediate and deep waters, which are less pronounced than in the surface water (Figs. 6 and 7). Overall, the
average vertical profile of $\varepsilon$Nd simulated in the SedRivDust experience is more consistent with the observed vertical profile (Fig. 5f), especially in the western basin where SeOnly and SedRiv largely overestimate the observations (Fig. 5d). This larger impact in the western basin is due to an input of dust with a low isotopic value in the southwestern basin ($\varepsilon$Nd = $\sim$ -14) while in the eastern basin, the dust input has a value more comparable with in situ observations ($\varepsilon$Nd = $\sim$ -12 for both). The Nd isotopic composition is largely affected by the $\varepsilon$Nd value in the atmospheric dusts, as shown by the Dust-CST and Dust-EWbasin
experiences, which largely underestimate $\varepsilon$Nd in the Mediterranean Sea as compared with in-situ data and the SedRivDust experience (Fig. 5).





## 4 Discussion

We simultaneously modeled $\varepsilon$Nd and [Nd] and explicitly represented all sources of Nd in the Mediterranean Sea by using a high-resolution coupled model (NEMO-MED12-PISCES), which includes the transport of Nd both by ocean dynamics and
particle scavenging. This modelling study confronted with observations provided new insights on the impact of the various sources of Nd on the $\varepsilon$Nd distribution and Nd budget in the Mediterranean Sea.

### 4.1 The Nd budget in the Mediterranean Sea

The SedRivDust experience provided the best agreements between simulation and in-situ observations, and allowed to derive a global Nd budget in the Mediterranean Sea. In this simulation, the total BE flux is estimated at $89.43 \times 10^6$ g(Nd)/yr, and
represents ∼90 % of the total flux of Nd into the Mediterranean basin. This estimation is relatively lower but comparable to the estimation of the net Nd release from the sediment by BE processes at the global scale (96.7 %, Arsouze et al., 2009). This result confirms that sediment deposited at the ocean boundaries (*i.e.* margins) should be considered as a major Nd source to the ocean and must be considered to simulate a realistic Nd oceanic cycle. Dissolved Nd input by rivers amounts to $3.6 \times 10^6$ g(Nd)/yr, which represents 3.7 % of the total Nd input to the Mediterranean Sea (2.3% at the global scale, Arsouze et al., 2009).
Finally, the atmospheric Nd input simulated here is $5.2 \times 10^6$ g(Nd)/yr, representing 5.3 % of the total Nd input. This flux is more than 5 times the global ocean one (0.96 % of the total Nd flux, Arsouze et al., 2009). Although significant, the relative contribution of the atmospheric source to the Mediterranean basin remains low compared to the BE input. Yet, it was essential to simulate more realistic Nd concentrations and isotopic compositions in the Mediterranean basin.

### 4.2 Evaluation of the impact of the external sources on the Nd Mediterranean Sea cycle

The first simulation, considering only sediment remobilization effects along the continental margin (*i.e.* boundary source, SedOnly experience), generates some characteristics of large-scale distribution of [Nd] and $\varepsilon$Nd and confirms sediment remobilization as the major source of Nd in the marine environment. It reinforces previous conclusions derived for the global ocean that BE is a major process in the Nd oceanic cycle. Nevertheless, on its own, sediment remobilization leads to a too-radiogenic isotopic Nd signature in the surface and intermediate waters as compared with in-situ observations, as was previously observed
by Ayache et al. (2016) and more recently by Vadsaria et al. (2019), both using more simplified modelling approaches. The results of this experience also generated low and homogeneous Nd concentrations in the surface waters that largely underestimated in-situ observations compared with the in-situ observations. This suggests that this unique source could not control alone the general distribution of [Nd] in the Mediterranean Sea.

Adding the dissolved river discharge in the second experience (SedRiv) is not significantly affecting the modelling results.
The main river systems of the Mediterranean basin (*i.e.* the Nile, Po and Rhone) are characterized by [Nd] of ∼34 ppm for the Nile, 25.77 ppm for the Rhone and 26.85 ppm for the Po river (Frost et al., 1986). They also display a wide range of Nd isotopic signatures, with an average $\varepsilon$Nd value of -10.2 for the Rhone, and more radiogenic Nd isotopic ratios for the Nile ($\varepsilon$Nd ∼ -4). The SedRiv experience generated $\varepsilon$Nd values very close to those of the SedOnly experience, as the river source has a





very similar isotopic signature to its neighboring continental margin. Moreover, the detectable impacts of river discharge on

the modeled Nd concentration were limited to the areas near the catchments of the main rivers. This is clearly visible for the surface waters in the vicinity of the Rhone river mouth (not shown). Overall, despite the consideration of the dissolved river input, [Nd] remained low and $\varepsilon$Nd too radiogenic in the surface waters.

The Saharan and middle east deserts located south and east of the Mediterranean Sea are sources of intense dust deposition events that affect the whole basin (Guerzoni et al., 1997). The Nd isotopic signatures of aerosols generated by these deserts

range from -9.2 in the eastern part of northern Africa to -16 in the central and western parts of northern Africa (Grousset and Biscaye, 2005; Scheuvens et al., 2013). Previous studies suggest that the $\varepsilon$Nd distribution at the near surface largely reflects river and aerosol inputs (Piepgras and Wasserburg, 1987; Jones et al., 2008; Arsouze et al., 2009). Including the atmospheric dust input in the SedRivDust experience greatly improved our simulation of the Nd Mediterranean cycle, with a more realistic simulation of $\varepsilon$Nd of the main water masses of the Mediterranean Sea, and corrected globally the too-radiogenic bias simulated

in the first two experiments (*i.e.* SedOnly and SedRiv). Even if the Nd isotopic compositions appear relatively too low in the Eastern Basin surface waters, they are more realistic in subsurface and deep-water masses. In addition, including aeolian dust added a significant amount of dissolved Nd to the surface water in all sub-basins, which greatly improved the simulated [Nd] concentrations toward the range of observed values. This increase in surface concentration also allows us to reproduce a more realistic average vertical profile, with a subsurface maximum detected in the observations. This signal corresponds to the

presence of a well-documented deep chlorophyll maximum (DCM) in the Mediterranean Sea (Cullen, 1982), whose associated primary production generates maxima in particle concentration (Annexed figure A5) where Nd molecule can be adsorbed and maintained in the water column.

Although the Nd flux associated with atmospheric deposition is much smaller than the BE flux (Tables 2 and 3), the results of our simulations show a significant impact of atmospheric dust on the Nd distribution in the whole basin. It seems paradoxical,

because dust input represents only 5.3 % of the total Nd input to the Mediterranean Sea. Sensitivity tests performed to better understand the influence of this source on the Nd oceanic cycle show that an increasing dust dissolution ratio is associated with more efficient scavenging, *i.e.* a more efficient transfer of tracer to the intermediate and deep waters and a lower concentration in subsurface waters (figure A3). Hence, the best model-data fit is obtained when applying a solubility of 2 % as suggested in many previously published studies (*e.g.,* Tachikawa et al., 1999; Lacan and Jeandel, 2001; Arraes-Mescoff et al., 2001; Arsouze

et al., 2009; Rempfer et al., 2011; Gu et al., 2019; Pöppelmeier et al., 2019). Considering various spatial distributions of Nd dust input led to similar results for surface water [Nd] as well as Nd vertical profiles (Fig. 5). This was however not the case for the Nd isotopic composition. Low dust $\varepsilon$Nd values characterize the western basin (-14) while they are more radiogenic in the eastern basin (Fig. 2f). As the magnitude of dust deposition is globally larger in the eastern than in the western basin (Fig. 2g), imposing constant values or averaged basin values (eastern and western basins) to the dust Nd isotopic composition led to

unrealistically low radiogenic values suggesting that considering the spatial distribution of the Nd isotopic composition in dust is essential. These results underlined that the modelled Mediterranean seawater Nd isotopic composition distribution is more sensitive than the modelled Nd concentration to the spatial characteristics of $\varepsilon$Nd in the atmospheric dust.





### 4.3 Internal cycle

The internal cycle also has a crucial role on the vertical distribution of Nd, reversible scavenging being the major process to
transfer the tracer into the deep layers (Nozaki et al., 1981; Siddall et al., 2005; Dutay et al., 2009; Arsouze et al., 2009). The
scavenging process is controlled by the partition coefficients and particle concentration. PISCES includes two categories of
particles, small POC particles with a low sinking speed (3m/d) and large particles with a larger sinking speed (50m/d). The
pool of large particles contains three types of particles: POC, $CaCO_3$, and BSi. There is currently not enough data to constrain
the partition coefficients for all these kinds of particles. We carried out sensitivity tests to assess the impact of this process on
the Nd distribution and try to reach a better agreement with the observations. The best compromise was found by increasing
only the partition coefficient for the small particles (*cf.* Figure A1). This result agrees with our previous modelling studies
on various tracers indicating that the internal cycle and vertical transport of Nd are mainly controlled by the small particle
pool (Arsouze et al., 2009), as was observed for $^{231}$Pa and $^{230}$Th (Dutay et al., 2009) and also in classical analytical studies
(Nozaki et al., 1981; Bacon and Anderson, 1982). The scavenging process also affects the vertical profile of the Nd isotopic
composition, lowering the Nd isotopic value in the water column. This study illustrates the role of scavenging in regulating the
vertical distribution of Nd in the Mediterranean basin. Our objective is not to estimate the most realistic values of K for our
simulation, as the simplification of our model could also be revisited and considered in the interpretation of our results. For
instance, the equilibrium hypothesis between the dissolved and particulate phases may not be always valid, especially for the
large particles, whose rapid sinking may not lead to equilibrium between the two phases.

Additionally, the concentration of particles is an important parameter to consider. An evaluation of the particle fields simu-
lated by PISCES at the global scale revealed that the small particles field ($POC_s$) is largely underestimated in deep water (up
to factor 4), and by a factor 2 for $CaCO_3$ concentration as compared to observations (Dutay et al., 2009; Hulten et al., 2018).
This issue highlights the need to consider more carefully the representation of the various particle fields, for the regional con-
figuration of the Mediterranean basin (NEMO-MED12/PISCES) as well, but this work is out of the scope of this preliminary
study.

### 5    Conclusions

This study proposes the first high-resolution simulation of both Nd concentration and isotopic composition in the Mediterranean
Sea, using a regional coupled dynamical/biogeochemical model and a reversible scavenging model to represent the reversible
exchange between the particulate and dissolved phases. We explicitly represented the main Nd sources from sedimentary
remobilization along continental margins (*i.e.* boundary exchange) as well as river discharge and atmospheric deposition at the
surface water. The objective was to determine and quantify the various sources involved in the Nd cycle, and to explore the
sensitivity to atmospheric dust deposition in the Mediterranean Sea.

It was confirmed that the sediment deposited on the margins is a major source of Nd to the ocean and is fundamental to
simulate a realistic Nd oceanic cycle. We estimated the BE flux at $89.43 \times 10^6$ g(Nd)/yr, which represents ~90 % of total
flux of Nd entering the Mediterranean basin, but is relatively lower than that estimated at the global scale (96.7 %). The rivers



provide $3.66 \times 10^6$ g(Nd)/yr, which represents 3.72 % of the total flow into the Mediterranean, compared with 2.3 % on the global scale. The flux of Nd from atmospheric dusts is estimated at $5.2 \times 10^6$ g(Nd)/yr, representing 5.3 % of the total Nd input, higher than in the global ocean, with only 0.96 % of the total Nd flux.

The impacts of river discharge on Nd concentration are limited to the areas near the catchments of the main rivers, *e.g.*, the
Rhone river, and lead to very low [Nd] in the surface water and too radiogenic $\varepsilon$Nd as compared with in situ data. Considering atmospheric dust inputs largely improved our simulation of the Nd oceanic cycle, with more realistic simulations of $\varepsilon$Nd in the main water masses of the Mediterranean Sea, and corrected the too-radiogenic bias simulated in our first two experiences (considering only the BE and river inputs), especially in the western basin. It also greatly improved the simulation of [Nd], generating values closer to the observed data, as well as a characteristic specific to the Mediterranean basin, a maximum
in subsurface associated to the DCM that was detected in the observations also. Based on the results of these sensitivity experiments, we suggest that the Nd cycle in the Mediterranean Sea is more impacted by atmospheric dust as compared to the global ocean due to its almost landlocked situation highly affected by dust deposition from the Sahara and Middle East. This work also suggests that $\varepsilon$Nd is more sensitive to the spatial distribution of Nd in the atmospheric dust, and confirmed that more in situ data and a better constraint of Nd fluxes from dissolved aeolian particles are necessary to improve our knowledge of the
cycles of Nd in the Mediterranean Sea.

Atmospheric dusts are only deposited in the surface layer (first model level) with a solubility ratio of 2 %, but uncertainty remains significant regarding their dissolution rates; a better constraint of this process would contribute to improve our constraint on the Nd cycle, especially in the Mediterranean basin where atmospheric deposition has a relatively greater influence. Additionally, more constraints on the K partition coefficient for the various types of particles will help to refine the representation
of the scavenging processes in the water column that control the transfer of the tracer into the intermediate and deep layers.

Clearly, more simulations, laboratory experiments and field observations are needed to better assess the influence of external sources (*e.g.* atmospheric dust, river, etc.) versus that of the internal cycle (*i.e.* scavenging/remineralisation). For instance, it would be useful to conduct similar analyses using other tracers (*e.g.* Sr, Si, etc.), or to use a more statistical analysis (*e.g.* TMM method) based on a multi-tracer approach. We demonstrated here the significant impact of atmospheric dusts on the Nd oceanic
cycle; it may be worth investigating in future studies their impact in other regions strongly affected by atmospheric input.

## 6 Code availability

The model used in this work is the free surfaceocean general circulation model NEMO (Madec and NEMO-Team., 2008) in a regional configuration called NEMO-MED12 (Beuvier et al., 2012b) (http://www.nemo-ocean.eu/).

## 7 Data availability

The data associated with the paper are available from the corresponding author upon request. All of the data used in this study were published by their authors as cited in the paper.





*Author contributions.* MA, JCD, TA contributed to the model development, simulations, and diagnostics. MA, JCD, KT, TA, CJ have been involved in the writing and revision of the manuscript.

*Competing interests.* The authors declare that they have no conflict of interest

*Acknowledgements.* The research leading to this study has received funding from the French National Research Agency ANR project Medsens.





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



**Table 1.** List of variables, units and presentation of all simulations used in this study

| Variable | Presentation | Unit |
|---|---|---|
| $\varepsilon_{Nd}$ | Nd isotopic composition | unit of $\varepsilon_{Nd}$ |
| [Nd] | Total Nd concentration | pmol kg$^{-1}$ |
| K | Equilibrium partition coefficient | - |
| $Nd_d$ | Nd dissolved concentrations | pmol kg$^{-1}$ |
| $Nd_p$ | Nd particulate concentrations | pmol kg$^{-1}$ |
| $C_p$ | Mass of particles per mass of water | Kg |
| $POC_b$ | Big particulate Organic Carbon | - |
| $POC_s$ | Small particulate Organic Carbon | - |
| $CaCO_3$ | Calcite | - |
| BSi | Biogenic Silica | - |
| litho | Lithogenic atmospheric dust | - |
| $Nd_T$ | Total concentration of Nd | pmol kg$^{-1}$ |
| $Nd_{ps}$ | Small particulate concentration | pmol kg$^{-1}$ |
| $Nd_{pb}$ | Big particulate concentration | pmol kg$^{-1}$ |
| S(NdT) | Source term of the Nd in the model | g yr$^{-1}$ |
| $\omega_s$ | Sinking velocities of small and big particles | m yr$^{-1}$ |
| $\omega_b$ | Sinking velocities of big particles | m yr$^{-1}$ |
| S($Nd_T$)sed | Source of BE (Boundary Exchange) | g yr$^{-1}$ |
| $F_{sed}$ | Source flux of sedimentary Nd to the ocean | g m$^{-2}$ yr$^{-1}$ |
| mask$_{margin}$ | Percentage of continental margin in the grid box | |
| S($Nd_T$)surf | Total source of Nd from river and from atmospheric dust | g yr$^{-1}$ |
| $F_{surf}$ | Nd flux of Nd from river discharge and from atmospheric dusts | g m$^{-2}$ yr$^{-1}$ |

| Simulations | Description |
|---|---|
| SedOnly | Considers sediment remobilization (i.e. Boundary Exchange) as the unique source of Nd. |
| SedRiv | Considers the dissolved fluvial material discharge in addition to sediment remobilization. |
| SedRivDust | Represents the three main inputs of Nd (i.e. boundary exchange + river discharge + atmospheric dust). |
| Dust-Cst | Same as RivSedDust run but with [Nd] ans $\varepsilon_{Nd}$ constant from atmospheric dust. |
| Dust-EWbasin | Same as RivSedDust run but with [Nd] ans $\varepsilon_{Nd}$ constant in atmospheric dust from eastern and western basins. |



**Table 2.** Main characteristics of source fluxes and equilibrium partition coefficients for each simulation.

| Experiences | | SedOnly | SedRiv | SedRivDust | Dust-Cst | Dust-EWbasin | Arsouze et al |
|---|---|---|---|---|---|---|---|
| | Sediment | $5.18 \times 10^6$ | $5.18 \times 10^6$ | $5.18 \times 10^6$ | $5.18 \times 10^6$ | $5.18 \times 10^6$ | - |
| Total quantity of Nd (g(Nd)) | River discharge | 0 | $9.12 \times 10^5$ | $9.12 \times 10^5$ | $9.12 \times 10^5$ | $9.12 \times 10^5$ | - |
| | Atmospheric dusts | 0 | 0 | $1.66 \times 10^6$ | $1.65 \times 10^6$ | $1.64 \times 10^6$ | - |
| | $K_{POMs}$ | $1.4 \times 10^8$ | - | - | - | - | $1.4 \times 10^7$ |
| | $K_{POMb}$ | $5.2 \times 10^4$ | - | - | - | - | $5.2 \times 10^4$ |
| Equilibrium partition coef. | $K_{BSi}$ | $3.6 \times 10^4$ | - | - | - | - | $3.6 \times 10^4$ |
| | $K_{CACO3}$ | $1.6 \times 10^5$ | - | - | - | - | $1.6 \times 10^5$ |
| | $K_{lith}$ | $4.6 \times 10^5$ | - | - | - | - | $4.6 \times 10^5$ |
| | Sediment | $89.4 \times 10^6$ | $89.4 \times 10^6$ | $89.4 \times 10^6$ | $89.4 \times 10^6$ | $89.4 \times 10^6$ | - |
| Total flux of Nd g(Nd)/yr | River discharge | 0 | $3.66 \times 10^6$ | $3.66 \times 10^6$ | $3.66 \times 10^6$ | $3.66 \times 10^6$ | - |
| | Atmospheric dusts | 0 | 0 | $5.3 \times 10^6$ | $5.25 \times 10^6$ | $5.2 \times 10^6$ | - |

**Table 3.** Estimation of the Nd flux from different sources in the Mediterranean Sea in comparison with the global ocean

| | Med Sea sum of flux in g((Nd)/yr) | % | Global ocean sum of flux in g((Nd)/yr) Arsouze et al., (2009) | % |
|---|---|---|---|---|
| Global flux Boundary Source | $89.4 \times 10^6$ | 90,7 | $1.1 \times 10^{10}$ | 96,7 |
| Dissolve fluvial material | $3.7 \times 10^6$ | 3,7 | $2.6 \times 10^8$ | 2,3 |
| Atmospheric dusts | $5.2 \times 10^6$ | 5,3 | $1.0 \times 10^8$ | 0,96 |
| Total | $98.3 \times 10^6$ | | $1.136 \times \times 10^{10}$ | |




**Figure 1.** Presentation of the main Nd oceanic modelling approach in the Mediterranean Sea. **(a)** modelling only the Nd isotopic composition ($\varepsilon$Nd), focused on the role of Boundary Exchange with the continental margin (on the first 540 m ) using a relaxing term (Ayache et al., 2016; Arsouze et al., 2007). **(b)** Explicitly representing the different sources of Nd to the ocean, *e.g.* sediment remobilisation (which implicitly represents the Boundary Exchange process), fluvial discharge, and atmospheric dust as done in Arsouze et al. (2009).





**Figure 2.** Boundary conditions and input maps applied to the model. **(a)** Nd concentration ([Nd], in $\mu$g/g) along continental margin determined by Ayache et al. (2016); squares, and hexagon represent in-situ data from the new global database of Nd provided by Blanchet (2019) . **(b)** Nd isotopic composition ($\varepsilon$Nd in $\varepsilon$Nd unit) along the continental margin determined by Ayache et al. (2016); squares, and hexagon represent in-situ data from the new global database of Nd provided by Blanchet (2019). **(c)** [Nd] of river runoff (in $\mu$g/g) from Ayache et al. (2016) with in-situ data from the new global database of Nd provided by Blanchet (2019). **(d)** $\varepsilon$Nd of river runoff (in $\varepsilon$Nd unit) presented in Ayache et al. (2016) with in-situ data from the new global database of Nd provided by (Blanchet, 2019). **(e)** [Nd] dust particle fields from the global database of (Blanchet, 2019; Robinson et al., 2021). **(f)** $\varepsilon$ Nd dust particle fields from the global database (Blanchet, 2019; Robinson et al., 2021). **(g)** Average deposition fluxes of dust (in g.m$^{-2}$) from the ALADIN Climate model (Nabat et al., 2015) ($10^6$ kg.m$^{-2}$.s$^{-1}$). **(h)** Runoff map prescribed by the NEMO-MED12 model (in $10^5$ g.m$^{-2}$.s$^{-1}$).



**Figure 3.** Nd concentration (in pmol/mol) for the surface level (0–200 m; left column), intermediate layer (250–600 m; middle column), and deep layer (600–3500 m; right column). Results from SedOnly experience, with only sediment remobilisation (**a, b, c**), SedRiv experience, with sediment and river input (**d, e, f**), and SedRivDust experience, with inputs from sediment, river and atmospheric dusts (**g, h, i**). Colour-filled dots represent in-situ observations from (Tachikawa et al., 2004; Vance et al., 2004; Henry et al., 1994; Dubois-Dauphin et al., 2017; Garcia-Solsona and Jeandel, 2020; Montagna et al., 2022). Both model and in-situ data use the same colour scale.





**Figure 4.** E-W vertical section of [Nd] (in pmol/kg) in the western Mediterranean basin from SedOnly **(a)**, SedRiv **(c)**, and SedRivDust **(e)**. E-W vertical section of [Nd] (in pmol/kg) in the western Mediterranean basin from SedOnly **(b)**, SedRiv **(d)**, and SedRivDust **(f)**; colour-filled dots represent in-situ observations from (Tachikawa et al., 2004; Vance et al., 2004; Henry et al., 1994; Dubois-Dauphin et al., 2017; Garcia-Solsona and Jeandel, 2020; Montagna et al., 2022). Both model and in-situ data use the same colour scale.



**Figure 5.** Upper panel: comparison of average vertical profiles of [Nd] (in pmol/kg) in the western basin **(a)**, eastern basin **(b)**, and whole Mediterranean Sea **(c)**, presenting model results against in-situ data from (Tachikawa et al., 2004; Vance et al., 2004; Henry et al., 1994; Dubois-Dauphin et al., 2017; Garcia-Solsona and Jeandel, 2020; Montagna et al., 2022). Lower panel: same as in upper panel for $\varepsilon_{Nd}$ (in $\varepsilon$ unit ).


**Figure 6.** Same as Figure 3 but for εNd (in εNd unit )



**Figure 7.** Same as Figure 4 but for $\varepsilon_{Nd}$ (in $\varepsilon$Nd unit ).





**Figure A1.** Left panel: E-W vertical section of [Nd] (in pmol/kg) in the entire Mediterranean Sea using the kd value from (Arsouze et al., 2008) (a), using the kd value from this study (b), and the difference between the two (c). Right panel: Comparison of average vertical profiles of [Nd] (in pmol/kg) in the whole Mediterranean Sea from the two experiences against in-situ data (black line) from (Tachikawa et al., 2004; Vance et al., 2004; Henry et al., 1994; Dubois-Dauphin et al., 2017; Garcia-Solsona and Jeandel, 2020; Montagna et al., 2022).





**Figure A2.** Left panel E-W vertical section of $\varepsilon_{Nd}$ (in $\varepsilon$ unit) in the entire Mediterranean Sea using the kd value from (Arsouze et al., 2008) (a), using the kd value from in this study (b), and the difference between the two (c). Right panel: Comparison of average vertical profiles of [$\varepsilon_{Nd}$ (in $\varepsilon$ unit) in the whole Mediterranean Sea from the two experiences against in-situ data (black line) from (Tachikawa et al., 2004; Vance et al., 2004; Henry et al., 1994; Dubois-Dauphin et al., 2017; Garcia-Solsona and Jeandel, 2020; Montagna et al., 2022).





**Figure A3.** Left panel E-W vertical section of [Nd] (in pmol/kg) in the entire Mediterranean Sea based on a 10% of dust solubility (a), a 2% of dust solubility (b), and the difference between the two (c). Right panel Comparison of average vertical profiles of [Nd] (in pmol/kg) in the whole Mediterranean Sea from the two experiences against in-situ data (black line) from (Tachikawa et al., 2004; Vance et al., 2004; Henry et al., 1994; Dubois-Dauphin et al., 2017; Garcia-Solsona and Jeandel, 2020; Montagna et al., 2022).



**Figure A4.** Left panel E-W vertical section of $\varepsilon_{Nd}$ (in $\varepsilon$ unit) in the entire Mediterranean Sea based on a 10% of dust solubility (a), a 2% of dust solubility (b), and the difference section (c). Right panel Comparison of average vertical profiles of $\varepsilon_{Nd}$ (in $\varepsilon$ unit) in the whole Mediterranean Sea from the two experiences against in-situ data (black line) from (Tachikawa et al., 2004; Vance et al., 2004; Henry et al., 1994; Dubois-Dauphin et al., 2017; Garcia-Solsona and Jeandel, 2020; Montagna et al., 2022).





**Figure A5.** E-W vertical section in the eastern Mediterranean basin (left panel), and comparison of average vertical profiles (right panel) for monthly mean climatological values of POC$_s$ (Small organic carbon Concentration), POC$_b$ (Big organic carbon Concentration), CaCO$_3$ (Calcite Concentration), and CHL total chlorophyll, for the western basin (red line) and eastern basin (blue line) in $\mu$ mol.l$^{-1}$.