# Peer review of "Neodymium budget in the Mediterranean Sea: Evaluating the role of atmospheric dusts using a high-resolution dynamical-biogeochemical model"

_Biogeosciences, 2022_

## Author Comment (AC1)

**Manuscript "***Neodymium budget in the Mediterranean Sea: Evaluating the role of atmospheric dusts using a high-resolution dynamical-biogeochemical model***"**

**Mohamed Ayache, Jean-Claude Dutay, Kazuyo Tachikawa, Thomas Arsouze, and Catherine Jeandel**

Dear Pr. A. Mazumdar

We would like to thank you for providing us the opportunity to revise our manuscript, and we would like to thank Reviewer 1 for taking the time and effort necessary to review the manuscript. We sincerely appreciate all valuable comments and suggestions, which helped us to improve the quality of the manuscript (ms hereafter).

**Color code**
Editor/Reviewer
Authors response
Change in the manuscript (ms)

**Anonymous Referee #1**

Ayache et al. present a revised implementation of the marine neodymium cycle in the high-resolution regional ocean model NEMO-PISCES for the Mediterranean Sea that forms the natural extension of previous work with less complex implementations. The authors now consider sediments, rivers, and dust as sources of Nd, and they assess their respective contribution and impacts via a number of sensitivity experiments. Similar to other recent studies, Ayache et al. come to the conclusion that sediments are the major source of Nd to the oceans also in the Mediterranean Sea. They further find that despite contributing only ~5% to the total Nd flux, dust plays a critical role for surface and intermediate Nd concentrations and isotopic compositions substantially improving the mismatch to observations.

Overall, the manuscript is well-written, but the figures require some improvements. The findings are in agreement with previous studies, and the new implementation in such a high-resolution regional ocean model will surely be a good test-bed for future investigations further elucidating the marine Nd cycle. However, some aspects remain unclear and require clarification as I outline below.

We thank the Reviewer for his/her interest in the manuscript and for highlighting the main points that should be considered for the revision. We have now significantly revised our manuscript, and we have restructured and rewritten it to give a clear general overview of the model and our modelling approach.

**Main points**

I think it would be helpful to give a brief description of the modern/simulated circulation in the Mediterranean Sea in the introduction, as not everyone may be familiar with the details of it. This would further allow for a more robust assessment of the applicability of Nd isotopes as faithful water mass tracer in the Mediterranean Sea later on that is currently missing.

We agree with this suggestion that not everyone may be familiar with the modern circulation in the Mediterranean Sea. Hence, we have added a brief description of Mediterranean Sea circulation in the revised ms (see below).

*"The Mediterranean is a concentration basin in which evaporation exceeds precipitation and river runoff. Warmer, fresher water enters at the surface from the Atlantic (Atlantic water – AW) through Gibraltar and colder saline water leaves below. The Levantine Intermediate Water (LIW) represents one of the main water masses of the Mediterranean Sea. It spreads throughout the entire Mediterranean basin at intermediate depths (between ~ 150 and 700 m) (Pinardi and Masetti, 2000). The LIW participates in the deep convection processes of the western Mediterranean deep water (WMDW) occurring in the Gulf of Lion and in the Adriatic sub-basin for the eastern Mediterranean deep water (EMDW) (Millot and TaupierLetage, 2005)."*

Why was no tuning of the Nd cycle parameters performed for this study, not even of unconstrained parameters of internal cycling? If this is too computationally expensive, this should be mentioned in the text also highlighting the downsides of an "un-tuned" Nd module with partly only poorly constrained parameters.

We agree with the referee that tuning the parameters of internal cycling of Nd cycle would give interesting information on the simulation and Nd oceanic cycle. However, internal Nd cycle depends on multiple quantities: particle fields (POCs, POCb, CaCO3 and BSi), partition coefficients (Kd) and settling velocities (w). Consequently, this issue requires to run many simulations which is not possible with our high-resolution model because it is too computationally expensive.

Moreover, there is currently not enough data to constrain the partition coefficients for all kinds of particles (POCs, POCb, CaCO3 and BSi). We have then carried out selected sensitivity tests to assess the impact of the processes on the Nd distribution and reach a better agreement with the observations. The best compromise was found by increasing only the partition coefficient for the small particles (*cf.* Figure A1).

We have tried to clarify and better justify the motivation of this choice in the revised manuscript.

*"We carried out sensitivity tests to assess the impact of this process on the Nd distribution and try to reach a better agreement with the observations. The best compromise was found by increasing only the partition coefficient for the small particles (cf. Figure A1). Internal Nd cycle depends on several parameters, particle fields (POCs, POCb, CaCO3 and BSi), partition coefficients (Kds) and settling velocities (w). Running many simulations for tuning these parameters is out of reach with the high computational cost of our high-resolution model."*

There appears to be no impact of riverine Nd to the Nd concentrations even at the surface. Even though in the text it is mentioned that there are small differences in the catchment areas, they are not visible in Fig. 3. I would have expected at least a visible difference close to the estuary of the Nile river. Since previous studies described the Nile as an important Nd source, I think the lack of a clear imprint thereof warrants a more detailed discussion of this.

The referee is right, the Nile has a very important impact on the Nd concentrations and the Mediterranean Sea circulation (e.g. during the Sapropel events for example). However, the construction of the Aswan High Dam had a major impact on the water discharge of Nile river (drastically reduced after 1964). River and runoff discharge forcing for the historical period are derived from the model of Ludwig et al. (2009) and the inter-annual data set of Vörösmarty et al. (1996). The detectable impacts of river discharge on modelled Nd concentration were limited to the areas near the catchment of the main rivers. This is clearly visible for the surface waters in the vicinity of the Rhone river mouth (see the difference between the red and green line in the surface water, Fig. R1). Finely, it is worth noting that the influence of river sediments is implicitly integrated in the BE term.

Changes were made in the text to clarify this point and we will introduce a new figure in appendix Fig. A6 (see below).

[Figure]

Figure R1: Comparison of the vertical profiles between in-situ data (from Henry et al. (1994)) and model output for: a) [Nd] (in pmol/kg), εNd in b).

*"The impact of river discharge on Nd concentration are limited to the areas near the catchment of the main rivers, i.e. the Rhone river. Almost all the main rivers presented*

*a significant discharge decreases (Ludwig et al., 2009) as a consequence of massive dam constructions (e.g. Aswan High Dam for Nile river). The detectable impacts of river discharge on modeled Nd concentration are limited to the areas near the catchment of the main rivers, i.e. the Rhone river where the impact is clear in the surface water (see Fig. A6). Finely, it is worth noting that the influence of river sediments is implicitly integrated in the BE term"*

Why does the dust flux generate a subsurface Nd maximum? If it is related to the subsurface production than this subsurface maximum should also emerge for the other simulations. Further, in line 283 it is mentioned that this subsurface maximum of experiment SedRivDust is also found in the observations, however, the depth profiles of Fig. 5 show a rather constant vertical profile with no pronounced maximum in the subsurface. In addition, as mentioned in line 359, it is paradoxical that such a small Nd flux by dust can increase the Nd concentration by so much. This paradox is not resolved in the text, and requires more in-depth investigation due to its great impact on the Nd cycle. In particular, it is very surprising to me that the riverine flux has virtually no impact on the surface and intermediate Nd concentrations, while the dust flux has a very large impact, while both sources are similar in magnitude (3.7% versus 5.3%).

We agree with this remark. It seems paradoxical, to see a such impact of atmospheric dusts while it represents only 5.3 % of the total Nd and that the riverine flux has virtually no impact on the Nd concentration in the surface and intermediate water of the Mediterranean Sea (as mentioned in the text line 358-366). We performed sensitivity test simulations to better understand the influence of these inputs on the Nd oceanic cycle, *i.e.* on the dissolution rates of particulate Nd from atmospheric dusts, and on the spatial distribution of [Nd] and εNd in atmospheric dust (cf. Section 2.5).

Effectively, considering atmospheric dust inputs largely improved our simulation of the Nd oceanic cycle, with more realistic simulations of εNd and [Nd] in the main water masses of the Mediterranean Sea, due to its almost landlocked situation highly affected by dust deposition from the Sahara and Middle East. It also generates a maximum in subsurface water, also detected in some in-situ data, as shown in Fig. R2, especially in the Ionian and Algerian sub-basins.

The high impact of atmospheric dust is explained by its injection that covers a large area over the Mediterranean Sea, where the deep chlorophyll maximum (DCM) is present, and constitutes a well-documented structure (Cullen, 1982) with high particle concentration (Annexed figure A5) where Nd can be adsorbed and maintained in the water column; while river flow and BE only affect coastal regions where DCM is not present.

[Figure]

**Fig. R2**: **(a)** Map of the NEMO-MED12 model domain and bathymetry with location of the main Mediterranean sub-basins and in-situ observation. The solid lines (in red) represent the trans-Mediterranean vertical section. Comparison of the vertical profiles of [Nd] (in pmol/kg) between in-situ data and model output for Levantine **b)**, Aegean **c)**, Ionian **d)**, Algerian **e)**, Gulf of Lion **f)**, and Alboran sub-basins.

On the other hand, the detectable impacts of river discharge on the modelled Nd concentration were limited to the areas near the catchments of the main rivers. This is clearly visible for the surface waters in the vicinity of the Rhone river mouth (as demonstrated in Fig. R1 and explained in the previous answer). The impact of river is relatively lower due to the reduction of river discharge.

It is difficult to assess the agreement between simulations and observations based on the figures alone. It would therefore be helpful to also provide a more objective measure, such as the root mean squared error, mean absolute error or another appropriate metric.

We would like to thank the referee for this suggestion. We performed several sensitivity tests to better understand how the internal cycle and the various external sources affect the Nd cycle in the Mediterranean Sea. Nevertheless, it is important to mention that the magnitudes and variations of Nd fluxes related to the partial dissolution of river particles and atmospheric dust bear a significant uncertainty. Hence, we didn't provide

a more quantitative measure because the model outputs are largely impacted by theses uncertainty in [Nd] from the different external sources. For instance, it would be useful to conduct similar analyses using other tracers (e.g. Sr, Si, etc.), or to use a more statistical analysis (e.g. TMM method) based on a multi-tracer approach.

**Specific points**

L26: I think it would be good to also cite more recent modeling work in this context (e.g., Gu et al., 2019; Pöppelmeier et al., 2022; Pasquier et al., 2022).

We would like to thank you for the mentioned references; we will introduce the references in the revised ms.

L64: I believe you are referring to Pöppelmeier et al. (2020), not Pöppelmeier et al. (2019).

Corrected

L75: Rempfer et al. (2011) and Gu et al. (2019) both also subtracted 70% of the total riverine Nd source not 30%.

We would like to thank the reviewer for this information, corrected in the revised ms

L152: 'adsorption onto particles' not 'into particles'.

Corrected

L161: Following not Flowing.

Corrected

L162: Do you mean that [Nd] and εNd are calculated offline?

We simulate the two $^{144}$Nd and $^{143}$Nd isotopes independently (simulates as two tracers) then we calculate total Nd concentration and εNd as a diagnostic parameter in the model.

Clarified in the revised ms.

"*Flowing Arsouze et al. (2009) we simulate the two $^{144}$Nd and $^{143}$Nd isotopes independently (simulates as two tracers) then we calculate total Nd concentration and εNd as a diagnostic parameter in the model.*"

L163: Mass-dependent fractionation is corrected for during measurement anyway.

Indeed.

L185: Typo – 'the the'.

Corrected

L200: The calculation of the sedimentary Nd flux remains unclear to me. How do you calculate the Nd flux from a bulk detrital concentration? Do you assume a constant and

uniform dissolution rate? Since this source flux is so important for the Nd cycle, a more thorough explanation is required.

We agree with the referee that the sedimentary Nd flux is so important for the Nd oceanic flux. To date, there is no estimation of the Nd flux from the sediment (*i.e.* the Boundary Source) in the Mediterranean Sea. Based on our modeling approach, we estimate the BE flux at 89.43 ×10$^6$ g(Nd)/yr for the whole Mediterranean basin (as presented in the ms). The only available estimation of the Nd Boundary Source flux was the global "missing flux" calculated by Tachikawa et al. (2003) and by Arsouze et al., (2003) using a coupled dynamical-biogeochemical model. We therefore use their value as a reference for our simulations, taking into account the percentage of the Mediterranean Sea as compared to the total surface of the global ocean.

Previous studies in the North Atlantic Ocean (Arsouze et al., 2007) and Mediterranean Sea (Ayache et al., 2016) taking into account the variability of the lithology of the margin sediments did not improve the simulations (Arsouze et al., 2007; Ayache et al., 2016). This requires more laboratory experiments, targeted on the issue of the nature of the sediments, Hence, we assume the sediment flux as geographically constant with a uniform dissolution rate as first approximation after many sensitivity simulations on the representation of this flux, the same assumptions were used in other modelling study (*e.g.* Arsouze et al., 2009). Then we computed the flux for both $^{144}$Nd and $^{143}$Nd by multiplying this sediment flux to the Nd concentration along the margin, based on the observed spatial distribution (Fig. 2a and 2b, in the submitted ms). We used a mask-margin which represent the percentage of continental margin in the grid box which, *i.e.* the proportion of the surface in the grid where the boundary exchange process occurs. The oceanic margin extension of the Mediterranean Sea has been chosen to be between 0 and ~540 m following the margin definition used to model the iron cycle in the Mediterranean Sea by Palmiéri (2014).

A sentence has been added to the text to make this perfectly clear in the revised manuscript.

"*To date, there is no estimation of the Nd flux from the sediment (i.e. the Boundary Source) in the Mediterranean Sea. Based on our modeling approach, we estimate the BE flux at 89.43 ×10$^6$ g(Nd)/yr for the whole Mediterranean basin (as presented above, see Table. 2). Previous study taking into account the variability of the lithology of the margin sediments in the North Atlantic Ocean (Arsouze et al., 2007) and the Mediterranean Sea (Ayache et al., 2016) did not improve the Nd simulations. This requires more laboratory experiments, targeted on the nature and reactivity of the sediments. Hence, we assume the sediment flux as geographically constant with a uniform dissolution rate as first approximation after many sensitivity simulations on the representation of this flux, the same assumptions were used in other modelling study (e.g. Arsouze et al., 2009).*"

Corrected

L240: Experiment not experience, here and elsewhere.

Donne

L 273: From Fig. 3 it appears that simulated Nd concentrations fit rather well to the observations in the deep layer and the factor of two difference mentioned in the text appears to be only present at intermediate depths.

We agree with the referee on this remarque, the factor of two is manly present in the layer of intermediate water as shown in Fig.3 and Fig. 5.

Text was changed in the revised ms for clarification.

"Nd concentration is increasing with depth in these two experiments, however, simulated concentrations only amount to roughly half the observed concentrations in intermediate  waters."

Done

Yes, it's true if we look only to the average vertical profile in the whole basin, there is an important difference in the western basin between the two first experiences (SedOnly and SedRiv) and the experiment with the atmospheric dusts (SedRivDust).

Clarified in the text.

"*Overall, the average vertical profile of εNd simulated in the SedRivDust experiments is more consistent with the observed vertical profile (Fig. 5f), especially in the western basin where SedOnly and SedRiv largely overestimate the observations (Fig. 5d) in the surface and intermediate water.*"

Changed.

L320. A brief comparison also to other global studies would be helpful to set these results into a better context.

We agree with the referee on the suggestion, we have introduced (*state*-of-the-*art*) the main modeling approach in section 1 (L 50-67) with the estimations of the Nd flux based on different molding approaches. We have discussed our estimations against the global study of Arsouze et al. (2009) because we have used a similar modeling approach giving a better comparability of our results.

Corrected

 Maybe give river Nd concentrations also in pmol/kg for better comparability to dissolved ocean concentrations.

Done

L360-363: I don't understand how a dust dissolution of 10% can lead to lower surface Nd concentration when the input Nd flux is five times higher. As mentioned in the text higher concentrations lead to more efficient scavenging, but since particle concentrations should remain the same between both experiments, the net effect on the dissolved Nd concentration should remain an increase not a decrease (albeit less that the factor of five).

As shown in Fig. A3 the experiment with a dissolution rate of 10 % give a relatively similar Nd concentration and less radiogenic water in the surface water (Fig. A4). The main difference is shown in the intermediate water as a consequence of a more efficient scavenging, *i.e.* a more efficient transfer of tracer to the intermediate and deep waters. This rapid transfer gives a lower concentration in subsurface waters (figure A3).

Figs. 2, 5: Please use the Greek epsilon character in the figures.

Done

Fig. 3: Please note whether these are averages over the mentioned depth intervals. Further, in the caption it is noted that the intermediate layer is from 250 to 600 m while in the text it is 200 to 600 m.

Clarified,  Fig. 3 display horizontal maps of Nd concentration averaged over the depth ranges surface (0- 200 m), intermediate (200-600 m) an deep water (600-3500 m).

Figs. 3, 4: Please use the same colorbar for both figures to allow for better comparability.

Changed

Fig. 6: The tick-labels of the colorbar appear to be rounded thus missing the digit after the comma.

Done

References

Gu, S. et al. Modeling Neodymium Isotopes in the Ocean Component of the Community Earth System Model (CESM1). J. Adv. Model. Earth Syst. 11, 624–640 (2019).

Pasquier, B. et al. GNOM v1.0: an optimized steady-state model of the modern marine neodymium cycle. Geosci. Model Dev. 15, 4625–4656 (2022).

Pöppelmeier, F. et al. Influence of elevated Nd fluxes on the northern Nd isotope end member of the Atlantic during the early Holocene. Paleoceanogr. Paleoclimatology 35, e2020PA003973 (2020).

Pöppelmeier, F. et al. Neodymium isotopes as a paleo-water mass tracer: A model-data reassessment. Quat. Sci. Rev. 279, 107404 (2022).

Rempfer, J., Stocker, T. F., Joos, F., Dutay, J. C. & Siddall, M. Modelling Nd-isotopes with a coarse resolution ocean circulation model: Sensitivities to model parameters and source/sink distributions. Geochim. Cosmochim. Acta 75, 5927–5950 (2011).

We would like to thank Referee #1 for the mentioned references; we will introduce the references in the revised ms.

**References**

Arsouze, T., Dutay, J. C., Lacan, F., and Jeandel, C.: Modeling the neodymium isotopic composition with a global ocean circulation model, Chemical Geology, 239, 165–177, https://doi.org/10.1016/j.chemgeo.2006.12.006, 2007.

Arsouze, T., Dutay, J.-C., Lacan, F., and Jeandel, C.: Reconstructing the Nd oceanic cycle using a coupled dynamical – biogeochemical model, https://doi.org/10.5194/bgd-6-5549-2009, 2009

Ayache, M., Dutay, J.-C., Arsouze, T., Révillon, S., Beuvier, J., and Jeandel, C.: High-resolution neodymium characterization along the Mediterranean margins and modelling of Nd distribution in the Mediterranean basins, Biogeosciences, 13, https://doi.org/10.5194/bg-13- 5259-2016, 2016.

Cullen, J. J.: The Deep Chlorophyll Maximum: Comparing Vertical Profiles of Chlorophyll a, Canadian Journal of Fisheries and Aquatic Sciences, 39, 791–803, https://doi.org/10.1139/F82-108, 1982.

Henry, F., Jeandel, C., Dupré, B., and Minster, J.-F.: Particulate and dissolved Nd in the western Mediterranean Sea: Sources, fate and budget, Marine Chemistry, 45, 283–305, https://doi.org/10.1016/0304-4203(94)90075-2, 1994.

Pinardi, N. and Masetti, E.: Variability of the large-scale general circulation of the Mediterranean Sea from observations and modelling: a review, Palaeogeography, Palaeoclimatology, Palaeoecology, 158, 153–173, 2000.

Millot, C. and Taupier-Letage, I.: Circulation in the Mediterranean Sea, The Handbook of Environmental Chemistry, 5, 29–66, doi:10.1007/b107143, 2005

Ludwig, W., Dumont, E., Meybeck, M., and Heussner, S.: River discharges of water and nutrients to the Mediterranean and Black Sea: Major drivers for ecosystem changes during past and future decades?, Prog. Oceanogr., 80, 199–217, doi:10.1016/j.pocean.2009.02.001, 2009.

Palmiéri, J.: Modélisation biogéochimique de la mer Méditerranée avec le modèle régional couplé NEMO-MED12/PISCES, http://www. 590 theses.fr/2014VERS0061, 2014.

Vörösmarty, C. J., Fekete, B. M., and Tucker, B. A.: Global River Discharge Database (RivDIS V1.0), International Hydrological Program, Global Hydrological Archive and Analysis Systems, UNESCO, Paris, 1996.

Tachikawa, K., Athias, V., and Jeandel, C.: Neodymium budget in the modern ocean and paleo-oceanographic implications, Journal of Geophysical Research, 108, 3254, https://doi.org/10.1029/1999JC000285, 2003.

---

## Author Comment (AC2)

**Manuscript "***Neodymium budget in the Mediterranean Sea: Evaluating the role of atmospheric dusts using a high-resolution dynamical-biogeochemical model***"**

**Mohamed Ayache, Jean-Claude Dutay, Kazuyo Tachikawa, Thomas Arsouze, and Catherine Jeandel**

Dear Pr. A. Mazumdar

We would like to thank you for providing us the opportunity to revise our manuscript, and we would like to thank Reviewer #2 for taking the time and effort necessary to review the manuscript. We sincerely appreciate all valuable comments and suggestions, which helped us to improve the quality of the manuscript (ms hereafter).

**Color code**
Editor/Reviewer
Authors response
Change in the manuscript (ms)

**Anonymous Referee #2**

This paper is an extension of Ayache et al. (2016) which intend to simulate epsilon Nd and its concentration in the Mediterranean Sea, using a regional dynamical-biogeochemical coupled model. In this paper, authors have considered Nd sources from benthic sediments, river discharge and atmospheric input to assess their relative contribution in the Nd Cycle for the Mediterranean Sea. Based on the modelling exercise, they have concluded that Sediments are dominant (almost 90 %) contributor of Nd to its oceanic cycle, with minor contribution from dust deposition and river input. While Nd contribution from atmospheric dust is low (~ 5%), it is very sensitive to Nd cycle and potentially important parameter to investigate in other regions which are strongly impacted by dust deposition.

I have thoroughly enjoyed reading results and discussion of this well drafted paper. This paper is of utmost importance for both atmospheric and Oceanic community and fits well within the scope of Biogeosciences. There are few typos in the draft, which I believe, will be taken care during proof reading stage. I recommend this paper for publication.

We warmly thank the reviewer for his overall encouraging comment concerning the utility of our study for the Bio-geoscience community.

Minor comments:

Line 63: can be reworded
We agree with the referee; this sentence was not very clear. Text was changed in the revised ms for clarification.

"*Ayache et al. (2021) used the simplified version of the εNd simulation proposed by Arsouze et al. (2007) to investigate with idealized hosing experiments in the IPSL-CM5 model the link between the intensification of the upper AMOC (Atlantic meridional overturning circulation) and the Mediterranean overflow*.

*Ayache et al. (2021) explores the impact of drastic changes in Mediterranean thermohaline circulation on the North Atlantic Circulation, using the simplified version of the εNd modelling approach (Arsouze et al. 2007) with idealized hosing experiments implemented in the IPSL-CM5 model.*"

Line 98: "…too-radiogenic…"   not clear

We meant that the simplified approach (including only the boundary exchange between sea water and continental margin, publish in Ayache et al., 2016) simulated a too-radiogenic isotopic composition of εNd, *i.e,* this approach overestimates the observed Nd isotopic composition.

Clarified in the revised ms

"*Nevertheless, this simplified approach yields too high (too radiogenic) eNd values compared to the modern Mediterranean Sea waters.*"

Line 138: C:N:P ratio is 122:16:1.. is it correct"?

Agreed. This was changed in the revised manuscript.

"*PISCES is a Redfieldian model where the C:N:P ratio used for plankton growth is fixed to 122:16:1*"

Line 141-143: Does smaller particle include Aeolian dust? It will be particularly important for open oceanic region as there is an enrichment of fine (clay) fraction in atmospheric deposition.

We totally agree with the referee about the role of Aeolian dust for open oceanic region. However, the small particle pool in the currents version of the biogeochemical model PISCES includes only particulate organic carbon (POCs, between 2 and 100 μm in size)

Our work highlights the need to consider more carefully the representation of the various particle fields in the biogeochemical model, and could be investigated in future studies.

Line 185: Typo "the"
Corrected

**References**

Arsouze, T., Dutay, J. C., Lacan, F., and Jeandel, C.: Modeling the neodymium isotopic composition with a global ocean circulation model, Chemical Geology, 239, 165–177, https://doi.org/10.1016/j.chemgeo.2006.12.006, 2007.

Arsouze, T., Dutay, J.-C., Lacan, F., and Jeandel, C.: Reconstructing the Nd oceanic cycle using a coupled dynamical – biogeochemical model, https://doi.org/10.5194/bgd-6-5549-2009, 2009

Ayache, M., Dutay, J.-C., Arsouze, T., Révillon, S., Beuvier, J., and Jeandel, C.: High-resolution neodymium characterization along the Mediterranean margins and modelling of Nd distribution in the Mediterranean basins, Biogeosciences, 13, https://doi.org/10.5194/bg-13- 5259-2016, 2016.

---

## Author Response (AR1)

**Manuscript "***Neodymium budget in the Mediterranean Sea: Evaluating the role of atmospheric dusts using a high-resolution dynamical-biogeochemical model***"**

**Mohamed Ayache, Jean-Claude Dutay, Kazuyo Tachikawa, Thomas Arsouze, and Catherine Jeandel**

Dear Pr. A. Mazumdar

We would like to thank you for providing us the opportunity to revise our manuscript, and we would like to thank Reviewer 1 and Reviewer 2 for taking the time and effort necessary to review the manuscript. We sincerely appreciate all valuable comments and suggestions, which helped us to improve the quality of the manuscript (ms hereafter).

**Color code**
Editor/Reviewer
Authors response

**Anonymous Referee #1**

Ayache et al. present a revised implementation of the marine neodymium cycle in the high-resolution regional ocean model NEMO-PISCES for the Mediterranean Sea that forms the natural extension of previous work with less complex implementations. The authors now consider sediments, rivers, and dust as sources of Nd, and they assess their respective contribution and impacts via a number of sensitivity experiments. Similar to other recent studies, Ayache et al. come to the conclusion that sediments are the major source of Nd to the oceans also in the Mediterranean Sea. They further find that despite contributing only ~5% to the total Nd flux, dust plays a critical role for surface and intermediate Nd concentrations and isotopic compositions substantially improving the mismatch to observations.

Overall, the manuscript is well-written, but the figures require some improvements. The findings are in agreement with previous studies, and the new implementation in such a high-resolution regional ocean model will surely be a good test-bed for future investigations further elucidating the marine Nd cycle. However, some aspects remain unclear and require clarification as I outline below.

We thank the Reviewer for his/her interest in the manuscript and for highlighting the main points that should be considered for the revision. We have now significantly revised our manuscript, and we have restructured and rewritten it to give a clear general overview of the model and our modelling approach.

**Main points**

I think it would be helpful to give a brief description of the modern/simulated circulation in the Mediterranean Sea in the introduction, as not everyone may be familiar with the details of it. This would further allow for a more robust assessment of the applicability of Nd isotopes as faithful water mass tracer in the Mediterranean Sea later on that is currently missing.

We agree with this suggestion that not everyone may be familiar with the modern circulation in the Mediterranean Sea. Hence, we have added a brief description of Mediterranean Sea circulation in the revised ms (See line 36-42 in the Author's track-changes file).

*Why was no tuning of the Nd cycle parameters performed for this study, not even of unconstrained parameters of internal cycling? If this is too computationally expensive, this should be mentioned in the text also highlighting the downsides of an "un-tuned" Nd module with partly only poorly constrained parameters.*

We agree with the referee that tuning the parameters of internal cycling of Nd cycle would give interesting information on the simulation and Nd oceanic cycle. However, internal Nd cycle depends on multiple quantities: particle fields (POCs, POCb, CaCO3 and BSi), partition coefficients (Kd) and settling velocities (w). Consequently, this issue requires to run many simulations which is not possible with our high-resolution model because it is too computationally expensive.

Moreover, there is currently not enough data to constrain the partition coefficients for all kinds of particles (POCs, POCb, CaCO3 and BSi). We have then carried out selected sensitivity tests to assess the impact of the processes on the Nd distribution and reach a better agreement with the observations. The best compromise was found by increasing the partition coefficient for the small particles only (*cf.* Figure A1).

*We have tried to clarify and better justify the motivation of this choice in the revised manuscript. See new section 4.3 in the revised ms and in the Author's track-changes file.*

*There appears to be no impact of riverine Nd to the Nd concentrations even at the surface. Even though in the text it is mentioned that there are small differences in the catchment areas, they are not visible in Fig. 3. I would have expected at least a visible difference close to the estuary of the Nile river. Since previous studies described the Nile as an important Nd source, I think the lack of a clear imprint thereof warrants a more detailed discussion of this.*

*We thank the referee for this comment, to which we agree. At first, it is worth noting that the influence of river sediments is implicitly integrated in the BE term, not treated with river water discharge. The Nile river water and particulate load had a very important impact on the Nd concentrations and the Mediterranean Sea circulation (e.g. during the Sapropel events for example). However, the construction of the Aswan High Dam drastically reduced the water discharge of Nile river after 1964. River and runoff discharge forcing for the historical period are derived from the model of Ludwig et al. (2009) and the inter-annual data set of Vörösmarty et al. (1996). The detectable impacts of river water discharge on modelled Nd concentration were limited to the areas near the catchment of the main rivers. This is clearly visible for the surface waters in the vicinity of the Rhone river mouth (see the difference between the red and green curves in the surface water, Fig. R1).*

*Changes were made in the text to clarify this point and we will introduce a new figure in appendix Fig. A6 (See line 376-383 and line 399-409 in the Author's track-changes file).*

[Figure]

**Figure R1:** *Comparison of the vertical profiles between in-situ data (from Henry et al. (1994)) and model output for: a) [Nd] (in pmol/kg), εNd in b).*

*Why does the dust flux generate a subsurface Nd maximum? If it is related to the subsurface production than this subsurface maximum should also emerge for the other simulations. Further, in line 283 it is mentioned that this subsurface maximum of experiment SedRivDust is also found in the observations, however, the depth profiles of Fig. 5 show a rather constant vertical profile with no pronounced maximum in the subsurface. In addition, as mentioned in line 359, it is paradoxical that such a small Nd flux by dust can increase the Nd concentration by so much. This paradox is not resolved in the text, and requires more in-depth investigation due to its great impact on the Nd cycle. In particular, it is very surprising to me that the riverine flux has virtually no impact on the surface and intermediate Nd concentrations, while the dust flux has a very large impact, while both sources are similar in magnitude (3.7% versus 5.3%).*

We agree with this remark. It seems paradoxical, to see a such impact of atmospheric dusts while it represents only 5.3 % of the total Nd and that the river water flux has virtually no impact on the Nd concentration in the surface and intermediate water of the Mediterranean Sea (as mentioned in the text line 410). We performed sensitivity test simulations to better understand the influence of these inputs on the Nd oceanic cycle, i.e. on the dissolution rates of particulate Nd from atmospheric dusts, and on the spatial distribution of [Nd] and εNd in atmospheric dust (cf. Section 2.5).

*Taking into account atmospheric dust inputs largely improved our simulation of the Nd oceanic cycle, leading to more realistic simulations of εNd and [Nd] in the main water masses of the Mediterranean Sea, due to its almost landlocked situation. It also generates a maximum in subsurface water, also detected in some in-situ data, as shown in Fig. R2, especially in the Ionian and Algerian sub-basins.*

The fundamental difference between the dust and river water flux is the fact that the atmospheric input contributes Nd to the whole Mediterranean surface water whereas the riverine influence is geographically localised. Spatial extension of the external source influence can be determined by the balance between water advection that transports the source signal from the source region and scavenging that removes added Nd from the water column. When the scavenging effect is dominant, the influence of external source would be localised. This could be the case for riverine inputs with visible influence is limited to river mouths. In contrast, the dust inputs affect the whole Mediterranean surface water including areas with low marine particle concentrations (Figure R3), allowing wider spatial extension of the source influence. This hypothesis is supported by the strong increase in total Nd amount in the Mediterranean Sea reflecting dust contribution (Table 2). The Nd added from dust source is vertically transported, leading to an increase in Nd concentration in the intermediate and deep waters, leading to better agreement with the field observation (Fig. R2). In addition, the contribution of unradiogenic Nd from dust corrects the positive bias of seawater Nd isotopic composition induced by strong BE influence (Figure 5).

*Although the deep chlorophyll maximum (DCM) is a well-documented structure (Cullen, 1982), subsurface maximum of dissolved Nd does not systematically appear in the Mediterranean Sea. It would be possible that simulated subsurface maximum reflects excessive scavenging in surface layer. Since we assume equilibrium scavenging, the scavenging efficiency is strongly controlled vertical distribution of POCs, POCb and CaCO3 (Figure A5). When the particle settling is fast, scavenging would not be at equilibrium state and the model overestimates the vertical transport. Moreover, scavenging constants may vary with water depths. Further studies will be required to examine these possibilities with more field observations.*

*While the atmospheric input is injected over a large area of the Mediterranean Sea, the deep chlorophyll maximum (DCM) is also present, and constitutes a well-documented structure (Cullen, 1982) with high particle concentration (Annexed figure A5). This provides sites where Nd can be adsorbed and maintained in the water column; while river flow only affect coastal regions where DCM is less stable. The parametrization of the vertical cycling (scavenging/remineralisation) considerably constrains the ability of the model to simulate the vertical profile of Nd concentrations, as shown in Fig. 5 the model underestimates the [Nd] of surface water as a consequence of an important transfer of tracer to the intermediate and deep water. For instance, the equilibrium hypothesis between the dissolved and particulate phases may not be always valid, especially for the large particles, whose rapid sinking may not lead to equilibrium between the two phases.*

[Figure]

**Fig. R2**: *(a) Map of the NEMO-MED12 model domain and bathymetry with location of the main Mediterranean sub-basins and in-situ observation. The solid lines (in red) represent the trans-Mediterranean vertical section. Comparison of the vertical profiles of [Nd] (in pmol/kg) between in-situ data and model output for Levantine b), Aegean c), Ionian d), Algerian e), Gulf of Lion f), and Alboran sub-basins.*

[Figure]

**Figure. R3:** Horizontal maps in surface water showing the monthly mean climatological values of POCs (Small organic carbon Concentration), POCb (Big organic carbon Concentration), CaCO3 (Calcite Concentration), and CHL total chlorophyll in μ mol.l⁻¹.

*Changes were made in the text to clarify this point and we will introduce a new figure in appendix Fig. A7 (See new section 4.2 in the Author's track-changes file).*

It is difficult to assess the agreement between simulations and observations based on the figures alone. It would therefore be helpful to also provide a more objective measure, such as the root mean squared error, mean absolute error or another appropriate metric.

We would like to thank the referee for this suggestion. We performed several sensitivity tests to better understand how the internal cycle and the various external sources affect the Nd cycle in the Mediterranean Sea. Nevertheless, it is important to mention that the magnitudes and variations of Nd fluxes related to the partial dissolution of river particles and atmospheric dust bear a significant uncertainty. Hence, we didn't provide a more quantitative measure because the model outputs are largely impacted by theses uncertainty in [Nd] from the different external sources. For instance, it would be useful to conduct similar analyses using other tracers (e.g. Sr, Si, etc.), or to use a more statistical analysis (e.g. TMM method) based on a multi-tracer approach.

Clarified in the revised ms (see the end of conclusion section, line 491-495 in the Author's track-changes file).

*Specific points*

**Specific points**

L26: I think it would be good to also cite more recent mmm work in this context (e.g., Gu et al., 2019; Pöppelmeier et al., 2022; Pasquier et al., 2022).

We would like to thank you for the mentioned references; we will introduce the references in the revised ms (see Section 1 line 78-83 in the Author's track-changes file).

PS, Gu et al., (2019) was already cited in the submitted ms.

L64: I believe you are referring to Pöppelmeier et al. (2020), not Pöppelmeier et al. (2019).

Indeed, we thank the referee for this remark (see line 74 in the Author's track-changes file).

L75: Rempfer et al. (2011) and Gu et al. (2019) both also subtracted 70% of the total riverine Nd source not 30%.

We would like to thank the reviewer for this information, corrected in the revised ms (see line 92-94 in the Author's track-changes file).

L152: 'adsorption onto particles' not 'into particles'.

Corrected (see line 171 in the Author's track-changes file).

L161: Following not Flowing.

Corrected (see line 180 in the Author's track-changes file).

L162: Do you mean that [Nd] and εNd are calculated offline?

We simulate the two $^{144}$Nd and $^{143}$Nd isotopes independently (simulates as two tracers) then we calculate total Nd concentration and εNd as a diagnostic parameter in the model.

Clarified in the revised ms (see line 181 in the Author's track-changes file).

L163: Mass-dependent fractionation is corrected for during measurement anyway.

Indeed.

L185: Typo – 'the the'.

Corrected.

L200: The calculation of the sedimentary Nd flux remains unclear to me. How do you calculate the Nd flux from a bulk detrital concentration? Do you assume a constant and uniform dissolution rate? Since this source flux is so important for the Nd cycle, a more thorough explanation is required.

We agree with the referee that the sedimentary Nd flux is so important for the Nd oceanic flux. To date, there is no estimation of the Nd flux from the sediment (*i.e.* the Boundary Source) in the Mediterranean Sea. Based on our modeling approach, we estimate the BE flux at 89.43 ×10$^6$ g(Nd)/yr for the whole Mediterranean basin (as presented in the ms). The only available estimation of the Nd Boundary Source flux was the global "missing flux" calculated by Tachikawa et al. (2003) and by Arsouze et al., (2009) using a coupled dynamical-biogeochemical model. We therefore use their value as a reference for our simulations, taking into account the percentage of the Mediterranean Sea as compared to the total surface of the global ocean.

Previous studies in the North Atlantic Ocean (Arsouze et al., 2007) and Mediterranean Sea (Ayache et al., 2016) taking into account the variability of the lithology of the margin sediments did not improve the simulations (Arsouze et al., 2007; Ayache et al., 2016). This requires more laboratory experiments, targeted on the issue of the nature of the sediments, Hence, we assume the sediment flux as geographically constant with a uniform dissolution rate as first approximation after many sensitivity simulations on the representation of this flux the same assumptions were used in other modelling study (*e.g.* Arsouze et al., 2009). Then we computed the flux for both $^{144}$Nd and $^{143}$Nd by multiplying this sediment flux to the Nd concentration along the margin, based on the observed spatial distribution (Fig. 2a and 2b, in the submitted ms). We used a mask-margin which represent the percentage of continental margin in the grid box which, *i.e.* the proportion of the surface in the grid where the boundary exchange process occurs. The oceanic margin extension of the Mediterranean Sea has been chosen to be between 0 and ~540 m following the margin definition used to model the iron cycle in the Mediterranean Sea by Palmiéri (2014).

A sentence has been added to the text to make this perfectly clear in the revised manuscript. See new section 2.4 (see line 229-235 in the Author's track-changes file).

We have added a new paragraph to the revised ms on the calculation of Nd residence time in the Mediterranean Sea (see change line 353-360 in the Author's track-changes file and new table 2).

L228: Typos in references.

Corrected

L240: Experiment not experience, here and elsewhere.

Done, thank you for this remark.

L 273: From Fig. 3 it appears that simulated Nd concentrations fit rather well to the observations in the deep layer and the factor of two difference mentioned in the text appears to be only present at intermediate depths.

We agree with the referee on this remark, the factor of two is manly present in the layer of intermediate water as shown in Fig.3 and Fig. 5.

Text was changed in the revised ms for clarification (see line 303 in the Author's track-changes file).

L290/291: Remove 'globally' here and elsewhere, since you only consider the Mediterranean.

We meat by globally over the whole Mediterranean Sea, and not the global scale.

L 300: The deep layer looks pretty much the same for all three experiments in Figs. 5f and 6

Yes, it's true if we look only to the average vertical profile in the whole basin, there is an important difference in the western basin between the two first experiences (SedOnly and SedRiv) and the experiment with the atmospheric dusts (SedRivDust).

Clarified in the text (see line 330 in the Author's track-changes file).

L313/338: Experiment not experience.

Changed.

L320. A brief comparison also to other global studies would be helpful to set these results into a better context.

We agree with the referee on the suggestion, and we have introduced (*state*-of-the-*art*) the main modeling approach in section 1 (line 58-83) with the estimations of the Nd flux based on different molding approaches. We have discussed our estimations against the global study of Arsouze et al. (2009) because we have used a similar modeling approach giving a better comparability of our results.

L332: Strike 'compared o in-situ observations'.

Corrected

L335-336: Maybe give river Nd concentrations also in pmol/kg for better comparability to dissolved ocean concentrations.

Done

L360-363: I don't understand how a dust dissolution of 10% can lead to lower surface Nd concentration when the input Nd flux is five times higher. As mentioned in the text higher concentrations lead to more efficient scavenging, but since particle concentrations should remain the same between both experiments, the net effect on the dissolved Nd concentration should remain an increase not a decrease (albeit less that the factor of five).

We would like to thank the referee for this remark. Since there was an error in the previous version of Fig. A3, we prepared a new version of Fig. A3 to better illustrate the impact of dust dissolution ratio (cf. the new Fig. A3 in the revised ms and line 412-418 in the Author's track-changes file).

[Figure]

**Fig. R4:** Left panel E-W vertical section of [Nd] (in pmol/kg) in the entire Mediterranean Sea based on a 10% of dust solubility (a), a 2% of dust solubility (b), and the difference between the two (c). Right panel Comparison of average vertical profiles of [Nd] (in pmol/kg) in the whole Mediterranean Sea from the two experiments (the experiment of 2% of dust solubility in blue and the experiment based on a 10% of dust solubility in green) against in-situ data (black line) from (Tachikawa et al., 2004; Vance et al., 2004; Henry et al., 1994; Dubois-Dauphin et al., 2017; Garcia-Solsona and Jeandel, 2020; Montagna et al., 2022).

As shown in Fig. A3 the experiment with a dissolution rate of 10 % give a globally higher Nd concentration and less radiogenic water in the surface water (Fig. A4). The main difference is shown in the intermediate and deep water and as a consequence of a more efficient scavenging, *i.e.* a more efficient transfer of tracer to the intermediate and deep waters.

Figs. 2, 5: Please use the Greek epsilon character in the figures.

Dune, changed to Nd IC (Nd Isotopic Composition)

Fig. 3: Please note whether these are averages over the mentioned depth intervals. Further, in the caption it is noted that the intermediate layer is from 250 to 600 m while in the text it is 200 to 600 m.

Clarified. Fig. 3 displays horizontal maps of Nd concentration (in pmol/mol) averaged over the depth ranges surface (0- 200 m), intermediate (200-600 m) an deep water (600-3500 m).

Figs. 3, 4: Please use the same colorbar for both figures to allow for better comparability.

Changed

Fig. 6: The tick-labels of the colorbar appear to be rounded thus missing the digit after the comma.

Done

References

Gu, S. et al. Modeling Neodymium Isotopes in the Ocean Component of the Community Earth System Model (CESM1). J. Adv. Model. Earth Syst. 11, 624–640 (2019).

Pasquier, B. et al. GNOM v1.0: an optimized steady-state model of the modern marine neodymium cycle. Geosci. Model Dev. 15, 4625–4656 (2022).

Pöppelmeier, F. et al. Influence of elevated Nd fluxes on the northern Nd isotope end member of the Atlantic during the early Holocene. Paleoceanogr. Paleoclimatology 35, e2020PA003973 (2020).

Pöppelmeier, F. et al. Neodymium isotopes as a paleo-water mass tracer: A model-data reassessment. Quat. Sci. Rev. 279, 107404 (2022).

Rempfer, J., Stocker, T. F., Joos, F., Dutay, J. C. & Siddall, M. Modelling Nd-isotopes with a coarse resolution ocean circulation model: Sensitivities to model parameters and source/sink distributions. Geochim. Cosmochim. Acta 75, 5927–5950 (2011).

We would like to thank Referee #1 for the mentioned references; we will introduce the references in the revised ms.

**Anonymous Referee #2**

This paper is an extension of Ayache et al. (2016) which intend to simulate epsilon Nd and its concentration in the Mediterranean Sea, using a regional dynamical-biogeochemical coupled model. In this paper, authors have considered Nd sources from benthic sediments, river discharge and atmospheric input to assess their relative contribution in the Nd Cycle for the Mediterranean Sea. Based on the modelling exercise, they have concluded that Sediments are dominant (almost 90 %) contributor of Nd to its oceanic cycle, with minor contribution from dust deposition and river input. While Nd contribution from atmospheric dust is low (~ 5%), it is very sensitive to Nd cycle and potentially important parameter to investigate in other regions which are strongly impacted by dust deposition.

I have thoroughly enjoyed reading results and discussion of this well drafted paper. This paper is of utmost importance for both atmospheric and Oceanic community and fits well within the scope of Biogeosciences. There are few typos in the draft, which I believe, will be taken care during proof reading stage. I recommend this paper for publication.

We warmly thank the reviewer for his overall encouraging comment concerning the utility of our study for the Bio-geoscience community.

Minor comments:

Line 63: can be reworded

We agree with the referee; this sentence was not very clear. Text was changed in the revised ms for clarification (see line 70-75 in the Author's track-changes file).

Line 98: "…too-radiogenic…"   not clear

We meant that the simplified approach (including only the boundary exchange between sea water and continental margin, publish in Ayache et al., 2016) simulated a too-radiogenic isotopic composition of εNd, *i.e,* this approach overestimates the observed Nd isotopic composition.

Clarified in the revised ms (see line 114-116 in the Author's track-changes file)

Line 138: C:N:P ratio is 122:16:1.. is it correct"?

Agreed. This was changed in the revised manuscript (see line 156 in the Author's track-changes file).

Line 141-143: Does smaller particle include Aeolian dust? It will be particularly important for open oceanic region as there is an enrichment of fine (clay) fraction in atmospheric deposition.

We totally agree with the referee about the role of Aeolian dust for open oceanic region. However, the small particle pool in the currents version of the biogeochemical model PISCES includes only particulate organic carbon (POCs, between 2 and 100 µm in size) Our work highlights the need to consider more carefully the representation of the various particle fields in the biogeochemical model, and could be investigated in future studies.

Line 185: Typo "the"
Corrected

**References**

Arsouze, T., Dutay, J. C., Lacan, F., and Jeandel, C.: Modeling the neodymium isotopic composition with a global ocean circulation model, Chemical Geology, 239, 165–177, https://doi.org/10.1016/j.chemgeo.2006.12.006, 2007.

Arsouze, T., Dutay, J.-C., Lacan, F., and Jeandel, C.: Reconstructing the Nd oceanic cycle using a coupled dynamical – biogeochemical model, https://doi.org/10.5194/bgd6-5549-2009, 2009

Ayache, M., Dutay, J.-C., Arsouze, T., Révillon, S., Beuvier, J., and Jeandel, C.: Highresolution neodymium characterization along the Mediterranean margins and modelling of Nd distribution in the Mediterranean basins, Biogeosciences, 13, https://doi.org/10.5194/bg-13- 5259-2016, 2016.

Cullen, J. J.: The Deep Chlorophyll Maximum: Comparing Vertical Profiles of Chlorophyll a, Canadian Journal of Fisheries and Aquatic Sciences, 39, 791–803, https://doi.org/10.1139/F82-108, 1982.

Henry, F., Jeandel, C., Dupré, B., and Minster, J.-F.: Particulate and dissolved Nd in the western Mediterranean Sea: Sources, fate and budget, Marine Chemistry, 45, 283– 305, https://doi.org/10.1016/0304-4203(94)90075-2, 1994.

Pinardi, N. and Masetti, E.: Variability of the large-scale general circulation of the Mediterranean Sea from observations and modelling: a review, Palaeogeography, Palaeoclimatology, Palaeoecology, 158, 153–173, 2000.

Millot, C. and Taupier-Letage, I.: Circulation in the Mediterranean Sea, The Handbook of Environmental Chemistry, 5, 29–66, doi:10.1007/b107143, 2005

Ludwig, W., Dumont, E., Meybeck, M., and Heussner, S.: River discharges of water and nutrients to the Mediterranean and Black Sea: Major drivers for ecosystem changes during past and future decades?, Prog. Oceanogr., 80, 199–217, doi:10.1016/j.pocean.2009.02.001, 2009.

Palmiéri, J.: Modélisation biogéochimique de la mer Méditerranée avec le modèle régional couplé NEMO-MED12/PISCES, http://www. 590 theses.fr/2014VERS0061, 2014.

Vörösmarty, C. J., Fekete, B. M., and Tucker, B. A.: Global River Discharge Database (RivDIS V1.0), International Hydrological Program, Global Hydrological Archive and Analysis Systems, UNESCO, Paris, 1996.

Tachikawa, K., Athias, V., and Jeandel, C.: Neodymium budget in the modern ocean and paleo-oceanographic implications, Journal of Geophysical Research, 108, 3254, https://doi.org/10.1029/1999JC000285, 2003.

---

## Author Response (AR2)

**Neodymium budget in the Mediterranean Sea: Evaluating the role of atmospheric dusts using a high-resolution dynamical-biogeochemical model**

Mohamed Ayache, Jean-Claude Dutay, Kazuyo Tachikawa, Thomas Arsouze, and Catherine Jeandel

**Color code**

Anonymous referee
Authors response

The authors thoroughly and convincingly answered all points raised by both reviewers and amended the main text accordingly, which significantly improved the clarity of the manuscript. There remain some small typographical errors (see below for some that I caught), but these can easily be amended at the proofreading stage. I therefore recommend publication of the manuscript.

We would like to thank you for taking the time and effort necessary to review the manuscript. We sincerely appreciate all valuable comments and suggestions, which helped us to improve the quality of the manuscript.

(Line reference correspond to track-changes version)
L61: Add tracer after quasi-conservative.

Done

L169: Change 'The observation ...' to 'Observations …'

Changed

L171: Change 'is redissolved' to 'redissolution'.

Changed

L213: Typo in 'established'.

Corrected

L239: 'unradiogenic' or 'less radiogenic' not 'non-radiogenic'.

Changed

L359: 'Upper limit' not 'higher limit'.

Changed